



# Assessment of Disdrometer Data Quality Control Methods for
# Precipitation Measurements Based on Wet-Bulb Temperature
Hyeon-Joon Kim[1], Sung-Ho Suh[2], Jongyun Byun[3], Changhyun Jun[4]
[1]Center of Oceanic and Meteorological Information, Pukyong National University, Busan, South Korea
[2]Flight Safety Technology Division, NARO Space Center, Korea Aerospace Research Institute, Goheung, South Korea
[3]Department of Civil, Environmental and Architectural Engineering, Korea University, Seoul, South Korea
[4]School of Civil, Environmental and Architectural Engineering, Korea University, Seoul, South Korea
*Correspondence to*: Changhyun Jun (cjun@korea.ac.kr)
**Abstract.** This study focuses on the reliability assessment of precipitation data calculated from drop size distribution (DSD)
based on disdrometer data observations according to wet-bulb temperature ($T_w$). Three distinct quality control (QC) methods
based on fall velocity were implemented and validated against measurements from tipping-buckets and weighing rain gauges
collected from January 2020 to February 2024. The analysis indicated that all QC methods exhibited high reliability
(correlation coefficient (CC) > 0.98) for rainfall conditions when $T_w$ was above 5 ℃, with a mean absolute percentage error
(MAPE) of approximately 8.5%. However, the precision of precipitation measurements exhibited a notable decline when $T_w$
was below 2 ℃, as indicated by a CC of less than 0.6 and MAPE exceeding 30%. This reduction in accuracy can primarily
be attributed to the outcomes of the QC methods, which rely on the falling velocity, given that raindrops and solid particles
were observed within the specified $T_w$ range. When considering the melting of snow particles at $T_w$ ranging from 0 ℃ to 2 ℃,
the CC approached 0.9, suggesting enhanced measurement reliability. The findings of this study indicate that $T_w$ is a more
effective variable than air temperature ($T_{air}$) for differentiating the precipitation types. This conclusion arises from the
observation that the fall velocity of hydrometeors does not reach the terminal velocity of raindrops, even within the $T_{air}$ range
of 1–5 ℃, coupled with the broad distribution of fall velocities. The DSD shape demonstrated stability across multiple QC
methods when $T_w$ was equal to or greater than 2 ℃. In contrast, considerable variations were observed at lower temperatures,
where particles with diameters ranging from 1 to 2 mm exhibited irregular distribution patterns at temperatures below 1 ℃.
These results suggest that DSD parameters should be derived from disdrometer data obtained under conditions where $T_w$ is
above 2 ℃ to ensure the reliability of the findings. This study provides critical insights for improving precipitation
measurement techniques and DSD analyses in regions with variable temperature conditions.



## 1 Introduction

Several factors influence the variability in precipitation development, including atmospheric water vapor content, vertical airflow intensity, and temperature and humidity distributions in the vertical profile (Lintner et al., 2017; Padullés et al., 2022). These factors can be categorized as climatological, geographical, or topographical. Climatological factors include alterations in atmospheric water vapor resulting from long-term temperature changes, developmental shifts associated with temperature variations in the upper and lower atmospheric layers, and thermodynamic effects linked to changes in land cover based on climatic characteristics (Dahlström, 2021; Lu et al., 2024). Geographical and topographical factors include the convergence of water vapor due to mountainous terrain, which facilitates vertical precipitation development (Insel et al., 2010; Lee et al., 2014; Kim et al., 2019), and the generation of vertical flow resulting from increased friction at low levels due to coastal topography (Du and Chen, 2019; Yao et al., 2021). Additionally, precipitation development can be influenced by the temperature differential between the sea and air, particularly when cold air from inland regions moves over water bodies during winter (Steenhurgh, 2020). Various environmental factors can influence the disparities in warm precipitation processes, such as collision-coalescence, evaporation, and accretion, which are contingent upon the vertical development of precipitation types, including stratiform, convective, and typhoon-related precipitation. Similarly, cold precipitation processes such as snow riming, melting, and ice crystal growth are affected by these environmental variables. These factors contribute to the development of diverse hydrometeors including rain, snow, and graupel, which are influenced by temperature variations (Maheskumar et al., 2018; Yi et al., 2021). The differences in the precipitation development processes ultimately lead to variations in the total precipitation observed at the surface. Therefore, it is crucial to acquire data that accurately reflects the microphysical characteristics of precipitation to enhance precipitation monitoring. Furthermore, analyses based on long-term observational data are essential to identify the universal characteristics that account for the temporal variability of precipitation.

The utilization of long-term observational data considerably reduces errors, mainly by rectifying inaccuracies in the observational data and eliminating outliers. Even among instruments that measure the same meteorological parameters, the threshold values for outlier removal may vary based on the installation conditions and surrounding environment. Additionally, biases in the observed values can arise owing to variations in the observation area and resolution, which are contingent on the type of instrument employed (Sypka, 2019; Segovia-Cardozo et al., 2021). Ground-based rain gauges can be categorized into two types based on their measurement method: tipping-buckets and weighing gauges. Although the tipping-bucket type demonstrates high accuracy in measuring rainfall, its efficacy in measuring snowfall during winter may be inferior to that of the weighing type because the observation value is recorded only when the precipitation in the bucket reaches a predetermined capacity (Savina et al., 2012; Kochendorfer et al., 2020). Comprehending the characteristics of observational instruments and the data they produce is imperative to ensure the reliability of the research findings derived from observational data.



The standard instruments used to observe precipitation include rain gauges and disdrometers. A rain gauge measures the total
precipitation accumulated over a specified time interval. In contrast, a disdrometer assesses the size and concentration of
precipitation particles, thereby enabling the determination of the precipitation intensity and type. The selection of an
appropriate type of rain gauge is contingent on specific observational objectives such as monitoring heavy rain, light rain, or
snow. Notable examples of disdrometers include the Particle Size and Velocity (PARSIVEL), Two-dimensional Video
Disdrometer (2DVD), Joss-Waldvogel Disdrometer, and Precipitation Occurrence Sensor System. Disdrometers compute the
size-specific concentration of particles, known as the Drop Size Distribution (DSD), by analyzing the variations in optical
intensity as the particles traverse the observation zone of the sensor. In contrast to data obtained from rain gauges,
disdrometer data offer a broader range of applications because they provide physical parameters, such as particle number
concentration and fall velocity, and morphological characteristics, such as oblateness.
A typical application of disdrometer data involves formulating Quantitative Precipitation Estimation (QPE) equations, which
are used in conjunction with remote sensing data such as radar observations. To derive rainfall information from remote
sensing data, researchers can leverage the correlation between rain rate and radar reflectivity values, an observational
variable in remote sensing, to measure variations in rainfall (Ji et al., 2019; Tang et al., 2024). Additionally, DSD
information obtained from disdrometer observations is instrumental in parameterizing microphysical schemes within
numerical weather prediction models (Yang et al., 2019; Iversen et al., 2021). Microphysical schemes can be categorized
into bin and bulk types (Hu and Igel, 2023). The bin scheme accurately simulates the distributional differences between
hydrometeor types by accounting for their size-dependent number concentration. However, this approach is limited by its
high computational demand and the need for substantial hardware resources during the simulation process. Consequently,
bulk schemes are predominantly employed in weather prediction models. This approach simulates microphysical processes
based on the relationships between particle diameter and concentration distributions for various hydrometeor types. The
DSD model considerably influences the quantitative differences in the estimated precipitation property outcomes. As the
reliability of the DSD model improves, so does the accuracy of the precipitation simulation and forecasting.
The DSD model used in the QPE and microphysical schemes of remote sensing encompasses various models, such as the
Marshall-Palmer, exponential (Marshall and Palmer, 1948), and gamma models (Ulbrich, 1983). The configuration of each
model is contingent on the specific parameters being analyzed, with the shape and slope variables in the DSD model varying
according to the concentration distribution based on the particle diameter (Smith, 2003; Liu et al., 2021). The DSD is
affected by several factors, such as the type of rainfall (Deo and Walsh, 2016), intensity of rainfall (Thomas et al., 2021), and
climatological and topographical characteristics of the region where precipitation occurs and develops (Kim et al., 2022).
Consequently, it is imperative to acquire DSD model parameters and reflectivity data by collecting highly reliable
observational data that accurately represent precipitation characteristics to enhance the precision of rainfall estimations and
simulations based on DSD. Furthermore, disdrometer data can be used to estimate rainfall erosivity (Serio et al., 2019).
Enhancing the accuracy of rainfall erosivity estimates can facilitate the assessment of the impact of rainfall on soil erosion
and serve as a foundation for developing countermeasures through spatial analysis and monitoring of soil erosion risk areas



using remote sensing data. Reliable precipitation observational data for estimating rainfall erosivity can aid in analyzing the
effects of erosion resulting from alterations in rainfall patterns due to climate change.
Various quality control (QC) methods for disdrometer data have been suggested to enhance the accuracy of derived
measurements (Kruger and Krajewski, 2002; Jaffrain and Berne, 2011; Raupach et al., 2015). QC approaches for
disdrometer data primarily rely on the falling velocity of raindrops. In the absence of a substantial wind influence or particle
collisions during descent, the fall velocity of a raindrop tends to increase with its diameter, ultimately reaching a terminal
velocity. Terminal velocity is achieved when the forces of air resistance and gravitational pull are in equilibrium, resulting in
no further particle acceleration. Studies have been conducted to determine the terminal velocities of raindrop particles (Atlas
et al., 1973; Beard, 1977; Brandes et al., 2002), which have led to the development of QC methods that use terminal velocity
measurements. Kruger and Krajewski (2002) elucidated the structural design and operational principles of a 2DVD system,
and noted that the recorded data indicated a fall velocity of approximately 400 m s$^{-1}$. However, this value is not feasible for
raindrops. To mitigate the impact of erroneous data (outliers) potentially arising from hardware malfunctions, inaccuracies in
data processing, and environmental conditions at the observation site, we employed a comparative analysis of the empirical
relationship of raindrops established by Atlas et al. (1973). Furthermore, recognizing that the disdrometer may either
underestimate or overestimate the fall velocity of precipitation particles influenced by the horizontal movement due to wind,
this study conducted QC by focusing exclusively on the vertical velocity measurements. Jaffrain and Berne (2011)
conducted a study to address the uncertainties associated with sampling observations from PARSIVEL disdrometers. They
argued that the collected precipitation data exhibit inherent variability and measurement errors attributable to the equipment
used, necessitating the development of a method to mitigate these issues and enhance data reliability. The authors proposed a
method for eliminating anomalous data, such as outliers (values that are not physically plausible), instances of particle
splashing (where the same particle is detected multiple times), and non-meteorological data. This preprocessing approach
effectively diminished the sampling uncertainty of various parameters, including rain rate.
Raupach et al. (2015) conducted a study using data from the PARSIVEL and 2DVD to establish a correction factor for
number concentration based on observations from the PARSIVEL disdrometer. The authors noted a tendency for
PARSIVEL to overestimate the number of small droplets measuring between 0.2 and 0.4 mm and larger particles measuring
2.4 mm or more. Furthermore, the measured fall velocity of larger droplets was lower than the actual terminal velocity.
Anomalous data can lead to DSD distortions, which can compromise the accuracy of precipitation measurements and radar-
based rainfall estimates. The focus of these studies was primarily on rainfall particles and it was determined that the
quantitative accuracy of rainfall estimates improved when the aforementioned QC methods were applied across various
environmental conditions.
Snow particles exhibit a variety of forms such as needles, dendrites, and granules, which are influenced by temperature and
humidity. These variations in shape arise from the specific conditions under which the particles form and develop, leading to
differences in their densities and fall velocities (Barthazy and Schefold, 2006; Vázquez-Martín et al., 2021). Furthermore,
snow particles are more susceptible to wind because of their lower density and larger surface area than raindrops.



Consequently, fall-velocity-based QC methods for eliminating non-meteorological particles (such as leaves, dust, and insects)
are limited in their effectiveness because they primarily target solid particles with low fall velocities. Given the diverse
shapes and fall speeds of snow particles, the mixing of raindrops and snow during precipitation events may lead to an
underestimation of errors when applying conventional disdrometer QC methods. Therefore, it is imperative to establish
objective criteria for differentiating rainfall and snowfall conditions to enhance the accuracy of rainfall analysis using
disdrometer data. Ding et al. (2014) emphasized the significance of accurately classifying precipitation types for surface
energy balance and hydrological process research. They aimed to develop a method for identifying precipitation types by
analyzing 30 years of observational data. Their investigation focused on the correlation between precipitation type and
various meteorological variables, including wet-bulb temperature ($T_w$), relative humidity ($RH$), and surface elevation. These
findings indicate that using $T_w$ as a reference variable for determining precipitation type is more reliable than relying on air
temperature ($T_{air}$). Furthermore, the proposed model, which incorporated $T_w$, demonstrated a determination accuracy
exceeding 88%.
This study aims to evaluate the quantitative accuracy of rainfall measurements obtained from a disdrometer in relation to
varying $T_w$ conditions. Furthermore, this study seeks to establish environmental criteria to ensure the reliability of the
parameters used in the DSD model by using long-term rainfall data collected through disdrometer observations. A
comparative analysis of the disdrometer data was performed using different QC methods to examine the discrepancies
between these methods under varying $T_w$ conditions.
**2 Data**
In this study, we evaluated the QC method applied to disdrometer data under varying precipitation conditions. To achieve
this, we collected and analyzed regional observational data that accounted for the environmental factors associated with
rainfall and snowfall. This study used data from a 2DVD installed at an observatory (Fig. 1) operated by the Weather Radar
Center of the Korea Meteorological Administration. The integrity of the 2DVD data was corroborated through comparisons
with measurements obtained from the tipping-bucket and weighing rain gauges. The analysis included observational data
collected between January 2020 and February 2024.



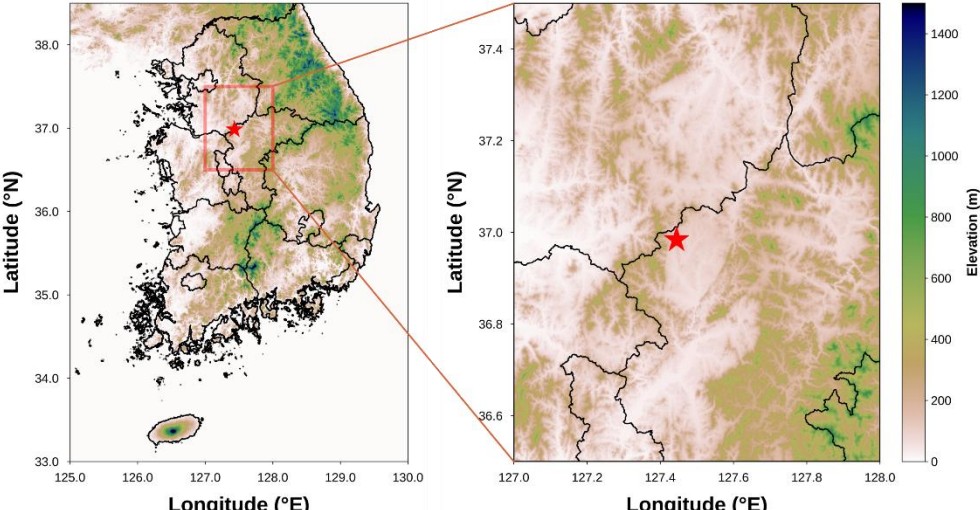

Figure 1: Location of ground observation station.

## 2.1 Disdrometer

The 2DVD (Kruger and Krajewski, 2002) used for the validity analysis of the disdrometer was an optical disdrometer developed by Joanneum Research. This instrument operates by projecting light through a bulb across a designated observation area and capturing the intensity of the transmitted light using a camera positioned on the opposite side (Fig. 2). When a particle, such as a raindrop, traverses the observation area (10 cm²) illuminated by the light sheet, its diameter is determined by analyzing the reduction in the intensity and width of the light during its passage. Furthermore, the system employs two cameras to observe the particles from orthogonal angles, allowing the fall velocity to be calculated based on the differential height of the light sheet in the two orientations and time taken for the particles to descend. The 2DVD's capability to acquire diameter and fall velocity data for individual particles offers superior temporal, dimensional, and velocity resolution compared to traditional disdrometer data, which typically provide channel-based information. The observational resolution of the camera was approximately 0.2 mm (512 pixels), making the particles smaller than the indistinguishable threshold (Grazioli et al., 2014). For quantitative validation using rain gauge data, the output time resolution was configured to one minute, with data classified at one-minute intervals.





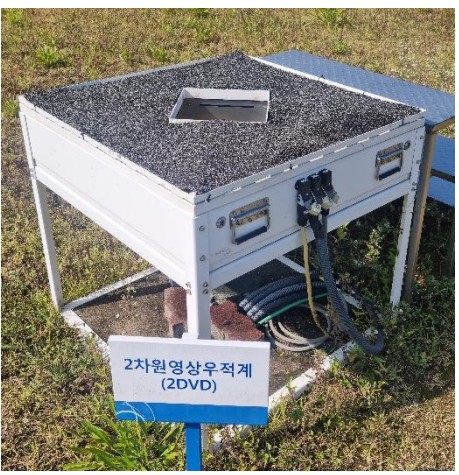

**Figure 2: Two-dimensional video disdrometer.**
**2.2 Rain gauge**
Precipitation can occur in liquid droplets and solid particles, such as snow and graupel, when temperatures are at or near 0 °C.
To validate the disdrometer data under $T_w$ conditions, an analysis was conducted using data from the tipping-bucket and
weighing-type rain gauges (Fig. 3). Each type of rain gauge offers an observational resolution of 0.1 mm and a temporal
resolution of 1 min. Both instruments were positioned within a 10 m radius of the 2DVD disdrometer.

(a) Tipping-bucket type          (b) Weighing type

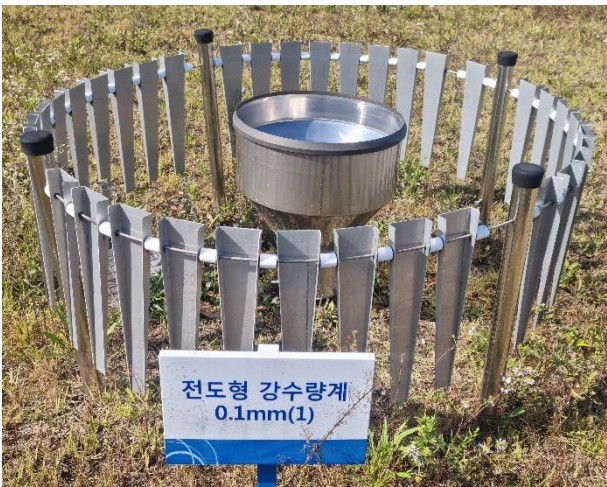
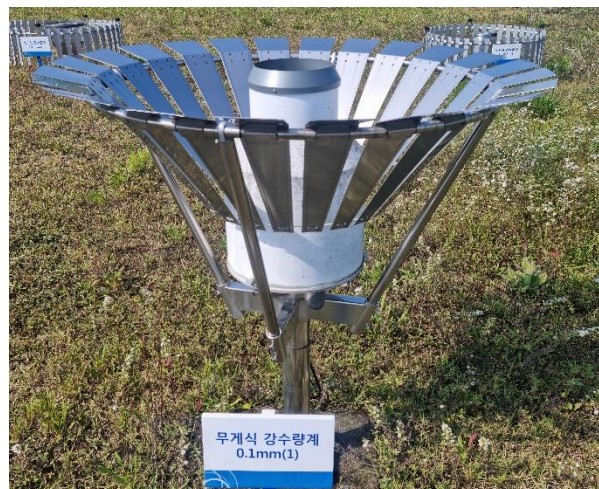

**Figure 3: (a) Tipping-bucket rain gauge (0.1 mm) and (b) Weighing rain gauge (0.1 mm).**





## 3 Methods


Ding et al. (2014) argued that precipitation types such as rain, snow, and sleet co-occur when the $T_{air}$ or $T_w$ approaches or
falls below 0 °C. They recommended using $T_w$ as a more effective criterion for distinguishing between types of precipitation
instead of relying solely on $T_{air}$. In this study, the temporal resolution of the temperature data differed from that of previous
studies, which employed different temporal resolutions. To facilitate objective verification of the applicability of $T_w$, $T_{air}$ and
$T_w$ were employed as criteria for classifying precipitation types, and a comprehensive analysis was conducted.

### 3.1 Pre-processing of disdrometer data


A common QC approach for disdrometer data involves excluding non-meteorological data by analyzing fall velocity. In
numerous studies, this QC process was implemented by establishing a threshold determined by the terminal velocity, as
indicated in Eq. (1).

$$|V_{measured} - V_{ideal}| < C \times V_{ideal} \tag{1}$$

where $V_{measured}$ and $V_{ideal}$ represent the observed particle fall velocity (in m/s) and empirical fall velocity (or terminal
velocity), respectively. Constant $C$ denotes the setting constant, which indicates the percentage of the terminal velocity. The
proportion of the removed particles may fluctuate based on the value of $C$. Numerous previous studies have provided
validation results using various setting constants. Studies that employed 2DVD data (Kruger and Krajewski, 2002; Thurai
and Bringi, 2005; Chang et al., 2009; Wen et al., 2018) predominantly adopted a setting constant of 0.4 (40%) during data
processing. Studies that employed PARSIVEL data for analysis frequently applied a setting constant of 0.6, accounting for
60% of the cases (Jaffrain and Berne, 2011; Friedrich et al., 2013; Ji et al., 2019; Kim et al., 2019). Given that previous
studies have encompassed various precipitation types, such as heavy rainfall, typhoons, orographic rainfall, and
thunderstorms, the established 40% and 60% QC conditions can be regarded as reliable preprocessing criteria for rainfall
events.
Raupach and Berne (2015) used data from a 2DVD instrument to derive correction factors for the drop-diameter channel in
the PARSIVEL dataset. The fall velocity filtering technique employed for the 2DVD and PARSIVEL data involved the
exclusion of particles exhibiting a terminal velocity exceeding 4 m s$^{-1}$, as shown in Eq. (2), those with a fall velocity below 3
m s$^{-1}$, as indicated in Eq. (3), and those larger than 7.5 mm, as shown in Eq. (4).

$$V_{measured} > V_{ideal} + 4 \tag{2}$$

$$V_{measured} > V_{ideal} - 3 \tag{3}$$

$$D > 7.5 \tag{4}$$

where $D$ (in mm) is the diameter of the drop (or particle). This study involved a comparative analysis of the outcomes
derived from the three QC methods based on fall velocity. Terminal velocity was determined using the equation established
by Atlas et al. (1973) (Eq. (5)).

$$V(D) = 9.65 - 10.3\exp(-0.6D) \tag{5}$$



Three QC methods were used to evaluate the research findings. Methods 1 and 2 are used for the ±40% and ±60% ranges of
terminal velocity, respectively, whereas Method 3 is based on the approach proposed by Raupach and Berne (2015).
As the temperature decreased, various hydrometeors intermingled, resulting in a gradual reduction in the proportion of
raindrops. Current QC methods are capable of eliminating low-density snow particles; however, to quantitatively compare
and validate rainfall measurements obtained from rain gauge observations, particles that exhibit velocities below the
threshold established for raindrops in each QC method are categorized as solid meteorological particles. In addition, analyses
were conducted under the assumption that the solid particles melted and transformed into raindrops. This method aims to
evaluate data from tipping-bucket rain gauges, which may exhibit diminished quantitative accuracy as the proportion of solid
particles increases, and facilitate quantitative comparisons of rainfall observations derived from disdrometer data by
implementing the QC method as the temperature decreases.
The equivalent-melted diameter ($D_{eq}$) at which a snow particle can transition into a raindrop while preserving its mass was
determined using Eq. (6), established by Delanoë et al. (2005). In this equation, $\rho(D)$ (g cm$^{-3}$) denotes the density of snow
particles as a function of their diameter, while $\rho_w$ (g cm$^{-3}$) denotes the density of water. The density of the snow particles was
computed based on the formula provided by Tiira et al. (2016) (Eq. (7)).

$$D_{eq} = \left( \rho(D) \big/ \rho_w \right)^{1/3} D \tag{6}$$

$$\rho(D) = 0.226 D^{-1.004} \tag{7}$$

**3.2 Raindrop size distribution**
The 2DVD data can be configured to correspond to user-defined diameter bin sizes, which in turn influence the
characteristics of the DSD output and the precision of the DSD model parameters (Marzuki et al., 2010). Consequently, this
study aims to facilitate the analysis of PASIVEL and 2DVD data for comparative purposes. To achieve this, 2DVD data
were processed using the diameter channel information derived from the PASIVEL data to compute the rain rate, number
concentration, and DSD model parameters. Detailed information regarding the diameter and velocity channels of the
PASIVEL data is provided in the appendices (Table A3-4). The rain rate ($R$, mm h$^{-1}$) is calculated using Eq. (8), which
incorporates the number concentration and fall velocity for each diameter. In determining the DSD model parameters after
the rain rate calculation, data from intervals where the rain rate was 0.1 mm h$^{-1}$ or greater were considered, thereby
minimizing the uncertainty associated with the DSD model. The gamma model, recognized for its reliability in representing
DSD characteristics, was selected for analysis. This model (Eq. (9)) is characterized by the shape parameter $\mu$ (Eq. (10)),
slope parameter $\Lambda$ (mm$^{-1}$) (Eq. (11)), and intercept parameter $N_0$ (mm$^{-1-\mu}$m$^{-3}$) (Eq. (12)).

$$R = \frac{6\pi}{10^4} \int_{D_{min}}^{D_{max}} D^3 N(D) V(D) dD \tag{8}$$

$$N(D)_{gamma} = N_0 D^\mu \exp(-\Lambda D) \tag{9}$$



$$\mu = \frac{(7 - 11\eta) - [(7 - 11\eta)^2 - 4(\eta - 1)(30\eta - 12)]^2}{2(\eta - 1)} \tag{10}$$

$$\Lambda = \left[\frac{M_2\Gamma(\mu + 5)}{M_4\Gamma(\mu + 3)}\right]^{1/2} = \left[\frac{M_2(\mu + 4)(\mu + 3)}{M_4}\right]^{1/2} \tag{11}$$

$$N_0 = \frac{\Lambda^{(\mu+3)}M_2}{\Gamma(\mu + 3)} \tag{12}$$


The DSD parameters were derived from the $n^{\text{th}}$ moment ($M_n$), as indicated in Eq. (13), along with the $\eta$ value, computed
based on $M_n$ as shown in Eq. (14).

$$M_n = \int_{D_{min}}^{D_{max}} D^n N(D)dD \tag{13}$$

$$\eta = \frac{<M_4>^2}{<M_2><M_6>} = \frac{(\mu + 3)(\mu + 4)}{(\mu + 5)(\mu + 6)} \tag{14}$$

**3.3 Wet-bulb temperature**
Data from an Automatic Weather Station (AWS) installed at the observatory were used to compute the $T_w$. The $T_{air}$ (in
degrees Celsius) and $RH$ (in percentages) values derived from the AWS observations were incorporated into the $T_w$ (in
degrees Celsius) calculation equation proposed by Stull (2011) (Eq. (15)) to determine the $T_w$ value. The temporal resolution
of $RH$, $T_{air}$, and $T_w$ was one minute, which was consistent with the temporal resolution of the disdrometer data.

$$T_w = T_{air}\,atan\left[0.151977(RH + 8.313659)^{1/2}\right] + atan(T_{air} + RH) - atan(RH - 1.676331) \tag{15}$$
$$+ 0.00391838(RH)^{\frac{3}{2}}\,atan(0.023101RH) - 4.686035$$

**4 Results**
**4.1 Comparison of rainfall by the disdrometer pre-processing method**
To validate the three QC methods employed for the disdrometer in this study, a comparative analysis was conducted between
the rainfall measurements obtained from the disdrometer and those recorded by rain gauges. This comparison utilizes hourly
accumulated rainfall data. Given that the QC methods for the disdrometer were specifically designed to address rainfall, the
variable $T_w$ was employed to differentiate between rainfall and snowfall, thereby facilitating the verification of rainfall
timing. Ding et al. (2014) argued that snow is infrequently detected when $T_w$ exceeds 5 °C. Figure 4 shows the distribution of
$T_w$ during the analysis period, specifically for instances when the hourly average $T_w$ was either above or below 5 °C. An
examination of the one-minute $T_w$ distribution during periods when the one-hour average $T_w$ was 5 °C or higher (Fig. 4a)
revealed a maximum $T_w$ of 26.2 °C, with the highest proportion of values exceeding 20 °C. Conversely, the proportion of
values falling below 5 °C was minimal, accounting for less than 5%. These findings suggest that it is feasible to delineate





rainfall periods using the hourly average $T_w$ as a reference when comparing hourly accumulated rainfall values. In contrast,
the distribution of one-minute $T_w$ during hours when the average $T_w$ was below 5 °C exhibited a broad range, with minimum
and maximum $T_w$ values exceeding 33 °C and a concentration of $T_w$ values around 0 °C. This observation indicates notable
variability in $T_w$ under 5 °C or lower, suggesting that the observational area encompasses environmental conditions
conducive to detecting diverse hydrometeors.

(a)

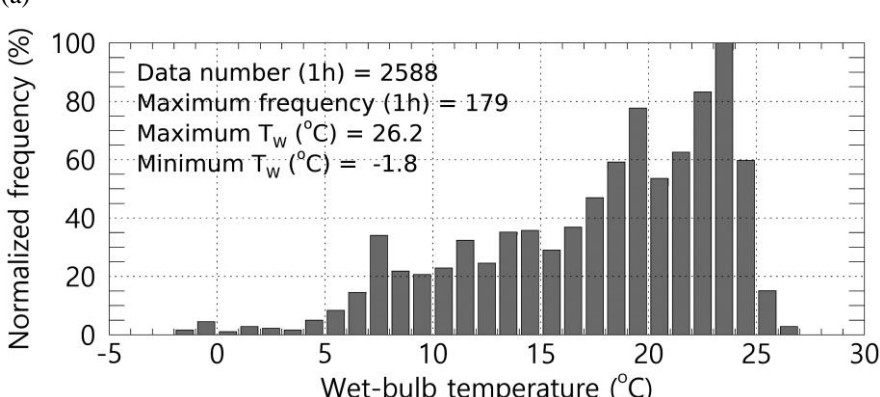

(b)

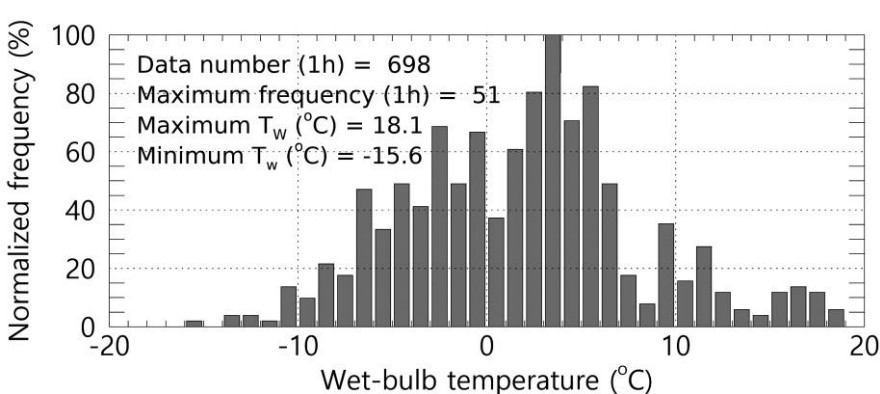

**Figure 4: Normalized frequency distribution of $T_w$ during the analysis period (when the average hourly $T_w$ is (a) $T_w \geq 5$ °C, (b) $T_w <$**
**5 °C).**
Figure 5-6 presents a comparative analysis of hourly rainfall measurements obtained from the tipping-bucket and weighing
rain gauge, specifically under conditions where the temperature ($T_w$) equals or exceeds 5 °C, alongside data from the 2DVD
observations. The results derived from the unprocessed raw data were analyzed to evaluate the impact of the QC procedures.
The findings indicated a strong correlation, exceeding 0.98, between the 2DVD and rain gauge measurements, with a
regression line slope of approximately unity. However, the raw data tended to overestimate the 2DVD-derived rainfall
estimates compared to the QC-processed results. This discrepancy in the overestimation of the 2DVD data can be attributed
to variations in the conditions under which particles are eliminated, which is contingent on the specific QC method
employed. Following the application of the QC methods, the mean absolute percentage error (MAPE) demonstrated an




overall reduction compared with the raw data, suggesting that all QC methods possess quantitative reliability for rainfall data,
with a maximum reduction of approximately 2.1%.

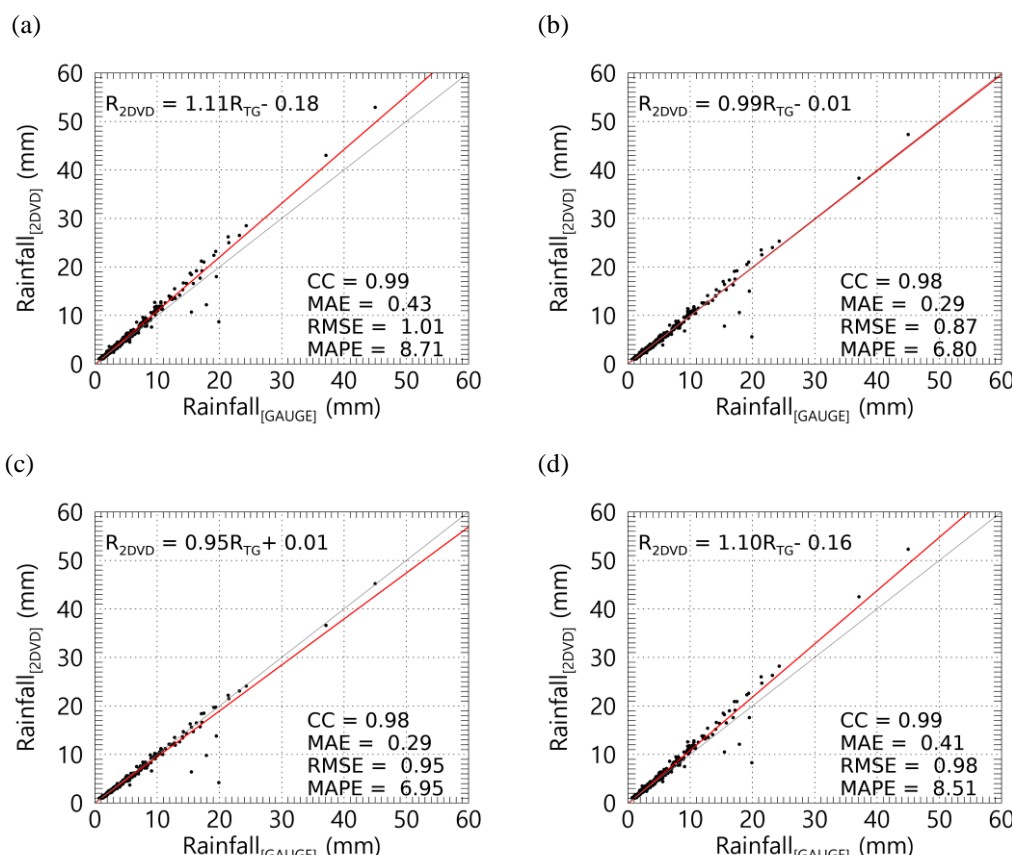

**Figure 5: Comparison of rainfall observed using the tipping-bucket rain gauge and 2DVD when $T_w \geq 5$ °C ((a) Unfiltered, (b)**
**Method 1, (c) Method 2, (d) Method 3). $R_{2DVD}$ and $R_{TG}$ denote the rainfall obtained from the 2DVD and a tipping-bucket rain**
**gauge, respectively.**








(a)                                    (b)





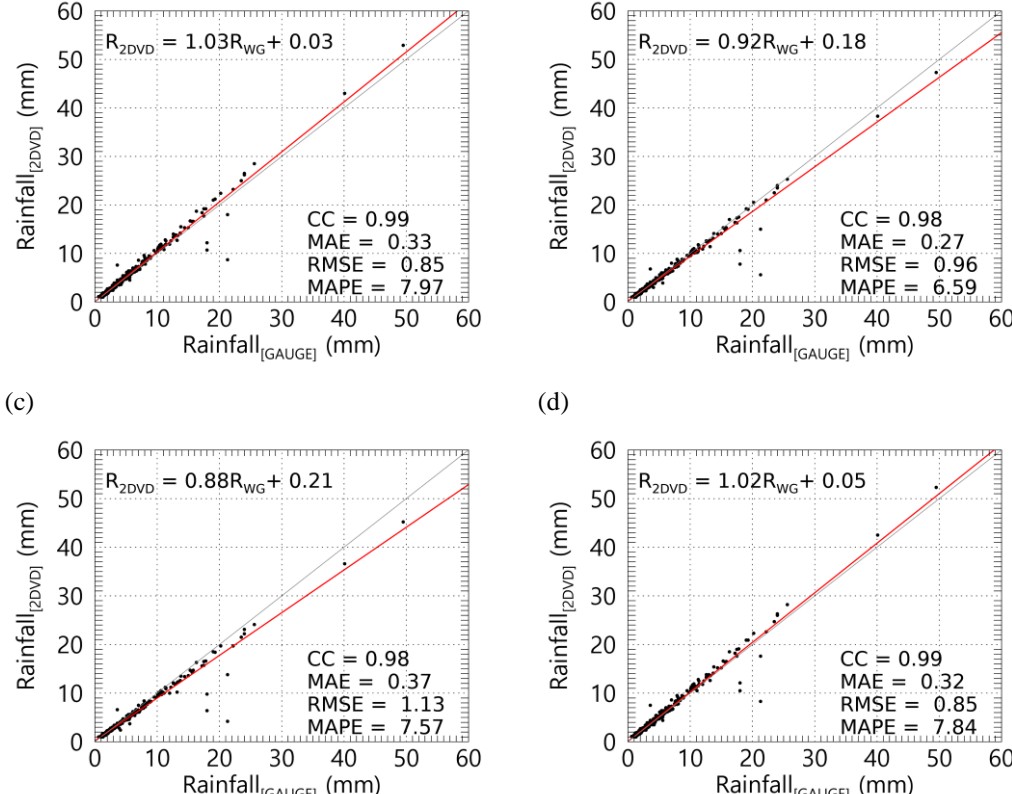

**Figure 6: Comparison of rainfall observed using the weighing rain gauge and 2DVD when $T_w \geq 5$ °C ((a) Unfiltered, (b) Method 1, (c) Method 2, (d) Method 3). $R_{WG}$ denotes the rainfall obtained from a weighing rain gauge.**

## 4.2 Fall velocity of particle by temperature and wet-bulb temperature

### 4.2.1 Fall velocity distribution at $T_{air}$ and $T_w$

Figure 7 shows the distribution of the fall velocity with the diameter of precipitation particles (raindrops) under varying conditions of $T_w$ and $T_{air}$. When the $T_w$ and $T_{air}$ ranged from -1 to 1 °C, the fall velocity distributions were relatively comparable. However, as the temperature exceeded 0 °C, the fall velocity for CH 4 to 18 increased under $T_w$ conditions, while the fall velocity for CH 19 to 23 increased under $T_{air}$ conditions (Fig. 7(a-b)). Notably, when the temperature rose above 1 °C, there was a notable increase in fall velocity; under $T_w$ conditions, the distribution approached the terminal velocity of raindrops. Conversely, for diameters in the CH 12 or a higher range, the fall velocity remained at approximately 5.5 m s$^{-1}$ despite increases in diameter. Under $T_{air}$ conditions, the fall velocity increased when temperatures were below 1 °C. However, it remained lower than that observed under $T_w$ conditions, with a broader distribution of fall velocities across the diameter channels (Fig. 7(c)). At 2 °C or higher temperatures, $T_w$ and $T_{air}$ conditions yielded fall velocity distributions that were close to the terminal velocity of raindrops, with an increasing trend in distribution as temperature increased (Fig. 7(d-f)).





However, under $T_{air}$ conditions, the fall velocity was notably low, remaining below 2 m s$^{-1}$ for diameters of 3 mm (CH 17) or
larger.

(a) -1 ℃ ≤ $T_w$ ($T_{air}$) < 0 ℃

(b) 0 ℃ ≤ $T_w$ ($T_{air}$) < 1 ℃

(c) 1 ℃ ≤ $T_w$ ($T_{air}$) < 2 ℃

(d) 2 ℃ ≤ $T_w$ ($T_{air}$) < 3 ℃

(e) 3 ℃ ≤ $T_w$ ($T_{air}$) < 4 ℃

(f) 4 ℃ ≤ $T_w$ ($T_{air}$) < 5 ℃

**Figure 7: Distribution of fall velocity by diameter channel based on $T_w$ (red) and $T_{air}$ (gray). The black solid line represents the**
**terminal velocity of rain drops proposed by Atlas et al. (1973).**



Figure 8 shows the variability in fall velocity with respect to the changes in temperature. Notably, despite $T_w$ and $T_{air}$
exhibiting similar numerical ranges, the distribution of fall velocity was considerably broader under $T_{air}$ conditions. When $T_w$
exceeds 3 ℃, the standard deviation across all diameter intervals remains low, approximately 1 m s⁻¹ or less. In instances
where $T_w$ ranges between 2 and 3 ℃, an increase in distribution is observed for diameters of 2.5 mm or greater, while the
standard deviation for diameters of 1 mm or more increases when $T_w$ is between 1 and 2 ℃. As temperature decreased, the
range of diameters exhibiting increased fall velocity variability progressively expanded. According to the findings under $T_{air}$
conditions, the standard deviation of fall velocity for diameters exceeding 1 mm begins to rise below 4 ℃, with values
surpassing 1 m s⁻¹ for diameters greater than 2 mm. The observation that when $T_{air}$ is between 2 and 3 ℃, the standard
deviation of fall velocity for diameters ranging from 3 to 4 mm is considerably increased, exceeding 2 m s⁻¹ and reaching up
to 4.5 m s⁻¹ is noteworthy. This broad fall velocity distribution suggests a mixture of various hydrometeors, complicating the
differentiation between rain and snow based solely on fall velocity. Consequently, subsequent analyses were conducted
using $T_w$ as the criterion for distinguishing between rain and snow.

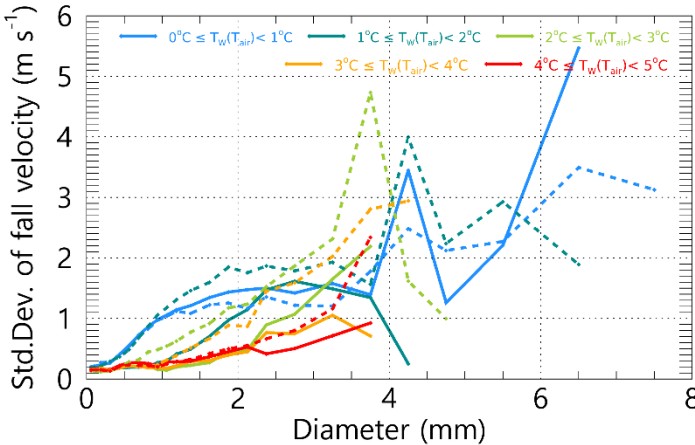

**Figure 8: Standard deviation of fall velocity by $T_w$ (solid line) and $T_{air}$ (dash line) range (1 ℃ interval).**
**4.2.2 Fall velocity distribution under rainfall condition**
Figure 9 shows the distribution of fall velocities by diameter, derived from data collected when the $T_w$ was at or above 5 ℃.
The central value of the fall velocity is consistent with the terminal velocity. This is within the range of fall velocities for
raindrops, as established by the three different QC methods based on the fall velocity. It is important to note that
precipitation particles (drops) may experience variations in their fall velocities owing to factors such as wind influence or
collisions with obstacles during descent. The findings presented in Fig. 9 suggest that the observatory's measurements were
not considerably affected by wind or obstacles, thereby confirming the reliability of the velocity observation data of the
disdrometer.





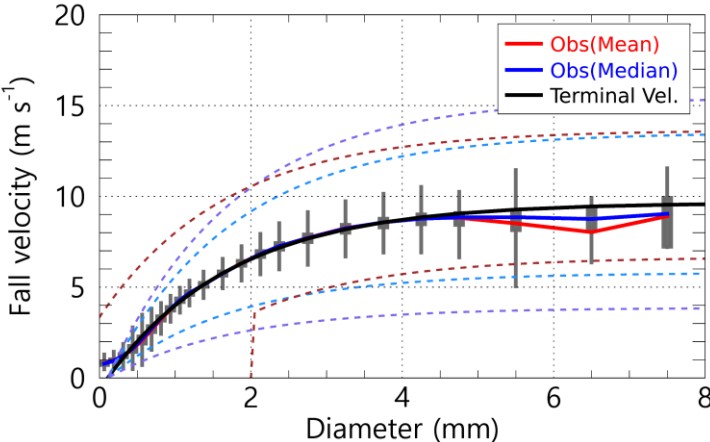

**Figure 9: Distribution of fall speed by diameter under conditions of $T_w \geq 5$ ℃, and effective fall velocity of raindrops by pre-processing methods (sky blue represents Method 1, purple Method 2, and brown Method 3).**

An analysis of the fall velocity corresponding to temperature intervals ($T_w$) of 1 ℃ revealed that when $T_w$ is at or above 2 ℃, the fall velocities correspond with those typically observed for raindrops. Conversely, at $T_w$ values between 1℃ and 2 ℃, particles with diameters of 2 mm or less fall within the raindrop velocity range; however, as the diameter increases to 2 mm or more, the fall velocity diminishes, stabilizing at approximately 5 m s$^{-1}$. Temperature conditions ($T_w$) may indicate a mixture of raindrops and snow particles. At temperatures below 1 ℃, the fall velocity of droplets with diameters of 4 mm or less decrease to approximately 3 m s$^{-1}$, exhibiting a low-velocity distribution of 5 m s$^{-1}$ or less across all diameter ranges. This distribution suggests a higher proportion of solid (snow) particles when $T_w$ is less than 1 ℃.

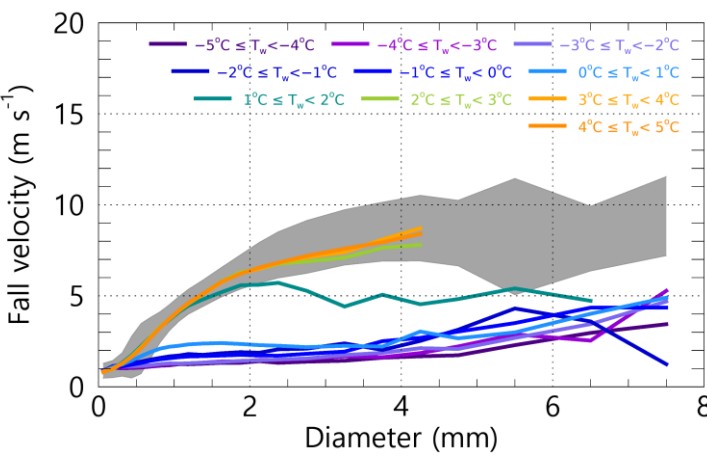

**Figure 10: Distribution of fall velocity by diameter in each $T_w$ range. The grey area in the figure represents the Q1 (25%) – Q3 (75%) for the fall velocity by diameter when $T_w \geq 5$ ℃.**





### 4.3 Accuracy of quantitative rainfall by wet-bulb temperature

Figures 11 and 12 show the outcomes of the comparative analysis and validation of rainfall measurements derived from the
QC method applied to each $T_w$ range, juxtaposed with the rainfall values obtained from a rain gauge. Figure 11 shows the
results of applying the QC method, which effectively filtered out all particles except for raindrops. In contrast, Fig. 12
depicts the assumption that the particles exhibiting low fall velocities are snow that melts and transforms into raindrops. The
verification metrics employed in this analysis included the Root Mean Square Error (RMSE), Mean Absolute Error (MAE),
MAPE, Correlation Coefficient (CC), slope ($a_1$), and intercept ($a_0$) derived from Eq. (14), which is the first-order regression
equation correlating rainfall measurements from the rain gauge and 2DVD.

$$R_{2DVD} = a_1 R_{Gauge} + a_0 \qquad (14)$$

The $a_1$ of the observed relationship indicates that when the $T_w$ exceeds 2 ℃, the value remains close to one before and
following the application of QC. However, as $T_w$ drops below 2 ℃, the value of $a_1$ either increases or decreases. A value of
$a_1$ greater than one suggests that the rainfall measurements derived from the 2DVD instrument tend to overestimate the
corresponding values obtained from the rain gauge observations. In contrast, a value of less than one indicates an
underestimation. Notably, in the absence of QC, $a_1$ increases to two or more at temperatures below 0 ℃, with the extent of
overestimation intensifying as $T_w$ decreases. This phenomenon is particularly evident when validated against a tipping-
bucket rain gauge, where values of two or greater were recorded at temperatures ranging from 0 to 1 ℃. This observation
may be attributed to the different operational principles of the various rain gauge types within the specified $T_w$ range (Fig.
11a). At $T_w$ below 0 ℃, the unfiltered data and Method 3 exhibit $a_0$ values exceeding one, while Method 2 and Method 3
present $a_1$ values below one. This discrepancy can be interpreted as a consequence of the varying quantities of preprocessed
particles. For $T_w$ values of 1 ℃ or higher, $a_0$ is observed to range between 0 and 1; however, as $T_w$ declines below 1 ℃, $a_0$
experiences a rapid increase. Method 3, which uses a smaller filter area for unfiltered particles and those with diameters of 2
mm or less, demonstrates $a_0$ values of 0.2 or higher, exceeding those of Methods 1 and 2 (Fig. 11b).
The CC decreases substantially in the $T_w$ range, whereas $a_0$ increases considerably (Fig. 11f). The RMSE and MAE were
recorded at low values of less than 0.3 mm and 0.2 mm, respectively, when $T_w$ was at or above 2 ℃; however, these errors
increased as $T_w$ decreased to 1 ℃ or lower, with the magnitude of errors following the order of Unfiltered, Method 3,
Method 1, and Method 2, which corresponds to the increasing trend of $a_1$. In the range of 0 to 2 ℃, the errors associated with
results validated by the tipping-bucket rain gauge were greater than those from the weighing rain gauge (Fig. 11(c-d)). The
MAPE exhibited its lowest error rate, below 20%, at temperatures between 3 and 4 ℃. It progressively increased with a
decrease in $T_w$, ultimately reaching values of approximately 30% or more at temperatures of 2 ℃ or lower.
Comparable findings were observed when it was assumed that the snow particles melted (Fig. 12), with an increase in error
as the temperature ($T_w$) dropped below 2 ℃. The distinction between melted and unmelted snow particles was demonstrated
using a weighing rain gauge as a verification tool. In scenarios where the melted state was disregarded at $T_w$ values lower
than 2 ℃, the variability in the MAPE and CC was substantial, which was contingent upon fluctuations in $T_w$. Conversely,



the variability decreased when the melted state was considered, and the CC remained elevated approximately at 0.8 or above.
The pronounced escalation in error within the 0–1 °C range can be attributed to precipitation detected by the 2DVD system
that was not captured by the Tipping-bucket rain gauge (Fig. A1(f) in Appendix).
Furthermore, the low volatility and high correlation observed in the verification results using the weighing rain gauge within
the $T_w$ range can be explained by incorporating raindrops and snow particles in the 0–1 °C range. By assuming melting of
snow particles, both forms of precipitation can be integrated into precipitation calculations. The weighing rain gauge
recorded precipitation values that accounted for the cumulative weight of all the raindrops and snow particles (Fig. A2(f) in
the Appendix).





























(a) a1

(b) a0

(c) RMSE

(d) MAE

(e) MAPE

(f) CC

**Figure 11: Quantitative comparison of rainfall from a rain gauge (The solid line represents the tipping-bucket and the dash line represents the weighing rain gauge) and 2DVD by $T_w$ (assuming that snow particles do not melt).**






**Figure 12: Quantitative comparison of rainfall from a rain gauge (The solid line represents the tipping-bucket and the dash line represents the weighing rain gauge) and 2DVD by $T_w$ (assuming that snow particles melt).**

## 4.4 Particle filter rate

Precipitation measurements obtained from the disdrometer were derived from raindrop (or snow particle) accumulation. The quantitative errors associated with these precipitation measurements were assessed by comparing the removal rates of raindrops (or snow particles) using the QC method. Figure 13 shows the removal ratios corresponding to the $T_w$ range and channel diameter. The two methods, Method 1 and Method 2, exhibit differences in the range of removal velocities based on particle diameter; specifically, Method 2 encompasses a broader spectrum of raindrop sizes compared to Method 1, leading to an increased removal rate when the $T_w$ is below 0 °C. Notably, the removal rate for Method 2 surpasses that of Method 1 at temperatures lower than -2 °C. Conversely, Method 3 did not allow the removal of particles smaller than 2 mm (as



indicated in CH 14), regardless of their low fall velocity, resulting in a consistent removal rate of 0%, irrespective of
variations in $T_w$. This suggests that the number of particles smaller than 2 mm may be greater in Methods 1 and 2.
Furthermore, the removal rate was lower when snow particles were assumed to have melted than when they had not melted.
Nonetheless, for particles with a diameter of 1 mm or less, the removal rate ranged from approximately 10% to 30% when $T_w$
exceeded 1 ℃, which appears to be attributable to the removal of particles exhibiting a fall velocity that exceeds the
raindrops.
































(a) Method 1

(b) Method 1 (melted)

(c) Method 2

(d) Method 2 (melted)

(e) Method 3

(f) Method 3 (melted)

**Figure 13: Particle filter ratio by diameter channel for $T_w$ according to the pre-processing method based on falling velocity.**







## 4.5 Contribution rate by particle diameter to precipitation intensity

Figure 14 shows the contribution rate of the number concentration by diameter to the precipitation intensity as derived from the disdrometer data. It is observed that when the $T_w$ exceeds 1 ℃, the contribution rate remains approximately 20% or lower across all diameters. Conversely, when $T_w$ is below 1 ℃, the concentration of particles measuring 3 mm (CH 15) or larger considerably influences the calculation of precipitation intensity. The contribution rate of 1.25 to 1.75 mm diameter (CH 11 to 13) decreased when the temperature was lower than 1 ℃. The decrease in the contribution rate of drops smaller than 3 mm and the increase in the contribution rate of larger drops was as a result of the decrease in the concentration of drops smaller than 3 mm through the QC process (Section 4.4), which increased the impact of relatively larger drops on the calculation of precipitation intensity. This phenomenon can be attributed to the direct proportionality of the precipitation intensity to $N(D)$ and $D^3$, indicating that an increase in the particle diameter substantially affected the results. After QC, a substantial increase in the contribution rate for a specific diameter may affect the precipitation intensity owing to a decrease in the concentration of drops in the diameter range with a lower contribution rate.

In scenarios where it is assumed that snow particles have melted, the diameter of these particles decreases, increasing the concentration of smaller particles. As a result, the contribution rate of diameter from approximately 0.5 to 1 mm (CH 5 to 10) increased. Notably, in Method 3, there was a minimal removal of particles smaller than 2 mm, which resulted in negligible differences between the scenarios that accounted for the melted state of snow particles and those that did not.





(a) Method 1

(b) Method 1 (melted)

(c) Method 2

(d) Method 2 (melted)

(e) Method 3

(f) Method 3 (melted)

**Figure 14: Precipitation contribution rate by diameter channel for $T_w$ using the pre-processing method based on falling velocity.**





### 4.6 Drop size distribution

### 4.6.1 Number concentration calculated by applying QC methods based on $T_w$

The precipitation intensity derived from disdrometer data is contingent on the number concentration; therefore, examining the distribution of the number concentration is imperative. Figure 15 shows the average distribution of the number concentration obtained by applying the QC method under varying $T_w$ conditions. Notably, when the temperature exceeded 2 ℃, the distributions yielded by all QC methods were comparable. Method 3 exhibited a relatively high concentration of small droplets measuring 1 mm or less, whereas the number of droplets measuring 1 mm or more showed minimal variation (Fig. 15h). This finding indicates that, at temperatures above 2 ℃, the output values remain consistent across different QC methods.

At temperatures ranging from 1 to 2 ℃, the distribution of particles exceeding 2 mm in size was distinctly differentiated according to the QC method employed. This finding suggests that the fall velocity of particles larger than 2 mm exhibits considerable variation within this temperature interval (Fig. 10). Conversely, at temperatures below 1 ℃, the distribution obtained through Method 3 displayed an anomalous pattern. This irregularity can be attributed to the failure of Method 3 to exclude snow particles smaller than 2 mm, leading to a higher concentration than that of the other QC methods. When considering the scenario in which particles are assumed to have melted, an increase in the concentration of water was observed for particles with a diameter of 1 mm or less when the $T_w$ was between 0 and 1 ℃, while the concentration of particles larger than 2 mm remained relatively unchanged.

In comparing scenarios where particles are assumed to have melted versus those that are not, no notable differences were observed at temperatures exceeding 1 ℃ (see Fig. 15g). However, within the temperature range of 0 to 1℃, there was an increase in the number of particles smaller than 1 mm. There was a similar distribution in the number of medium and larger particles (1 mm or more). As the $T_w$ progressively decreased below 0 ℃, the disparity in the number concentration of particles larger than 1 mm became more pronounced (Fig. 15(a-e)).







**Figure 15: Average number concentration distribution for $T_w$ using pre-processing methods.**





### 4.6.2 Difference in the number concentration based on the gamma model

The change in the shape of the number concentration within the observed data has implications for DSD model parameters. The notable discrepancy between the observed number concentration and that derived from the model parameters raises concerns regarding the reliability of the DSD model.

Figures 16 and 17 show the variance between the observed number concentration and that predicted using the gamma model. When all QC methods were implemented, the MAPE remained below 60% across all diameter ranges at temperatures exceeding 2 °C. However, as the $T_w$ fell below 1 °C, the discrepancy for diameters greater than 0.6 mm (CH 7) escalated to over 70%. At 2 °C or higher temperatures, the gamma distribution overestimated the concentration of particles smaller than 1 mm while underestimating those larger than 1 mm. Nonetheless, the extent of under- or over-simulation by gamma distribution remained below 50% across all diameter intervals.

When $T_w$ was below 1 °C, assuming that the snow particles had melted, the error rate in simulating the concentration of particles smaller than 1 mm (CH 8) diminished (Fig. 16(b, d)). Concurrently, the percentage bias (PBAIS) for particle diameters less than 1 mm decreased, approaching a value near zero (Fig. 17(b, d)). This phenomenon can be attributed to the application of the QC method under subzero conditions, which led to an overestimation of the gamma distribution for diameters of 1 mm or less because of the increased influence of smaller particles resulting from the exclusion of larger particles exceeding 3 mm. Conversely, this resulted in an underestimation of the gamma distribution for diameters larger than 3 mm.



(a) Method 1

(b) Method 1 (melted)

(c) Method 2

(d) Method 2 (melted)

(e) Method 3

(f) Method 3 (melted)

**Figure 16: MAPE for diameter and wet-bulb temperature using the pre-processing method**





(a) Method 1

(b) Method 1 (melted)

(c) Method 2

(d) Method 2 (melted)

(e) Method 3

(f) Method 3 (melted)

**Figure 17: PBAIS for diameter and wet-bulb temperature using the pre-processing method**



## 5 Conclusion


This study employed data collected from a 2DVD disdrometer in conjunction with traditional rain gauges to assess the
precipitation measurements derived from the disdrometer under $T_w$ conditions and to evaluate the reliability of the DSD
model.
The precipitation estimates derived from the QC methods employed in this study exhibited CC ≥ 0.98 when juxtaposed with
precipitation measurements obtained from rain gauges in an environment with $T_w$ of 5 ℃ higher. The MAPE was
approximately 8.5%. In contrast to scenarios in which the QC method was not implemented, the propensity for the 2DVD
data to be overestimated diminished, and the overall error rate reduced. These findings indicate that all QC methods
demonstrated high reliability under rainfall conditions.
When $T_{air}$ and $T_w$ dropped below 1 ℃, there was a notable reduction in the fall velocity of precipitation particles, which
became concentrated within a velocity range of approximately 0.5 to 3 m s$^{-1}$. Moreover, the ratio of snow particles to
raindrops considerably increased at temperatures below 1 ℃ for $T_{air}$ and $T_w$. This observation is consistent with the findings
of Ding et al. (2014), which indicated that the proportion of rain was less than 30% at temperatures below this threshold.
Conversely, when $T_{air}$ ranged from 1 to 3 ℃, the distribution of fall velocities exhibited a broader range compared to
conditions where $T_w$ was between 1 and 3 ℃, resulting in a greater disparity with the terminal velocity of raindrops.
Consequently, it is justifiable to use $T_w$ as a reference parameter to differentiate between types of precipitation using
disdrometer data.
The fall velocity of particles exceeding 2 mm in diameter decreased within the 1 to 2 ℃ temperature range for $T_w$.
Conversely, at $T_w$ below 1 ℃, the observed results were outside the effective fall velocity range typically associated with
raindrops. Furthermore, as $T_w$ decreased below 2 ℃, the quantitative error, as measured by the correlation of disdrometer-
based precipitation data, increased. These results can be attributed to the fact that the filter ratio for particles measuring 3
mm or less escalated to 30% or higher when $T_w$ was below 2 ℃. Within this $T_w$ range, it is plausible to regard raindrops and
solid particles as intermixed; consequently, implementing QC methods appropriate for conventional rainfall scenarios may
lead to inaccuracies. However, when snow particles were assumed to have undergone melting, the correlation approached
0.9, even within the temperature range of 0 to 1 ℃, and the variability of the error decreased. These findings indicate that the
reliability of precipitation calculations can be maintained, even in scenarios where raindrops and snow particles coexist at
temperatures between 0 and 2 ℃, provided that an appropriate density for snow particles is used. Additionally, it is
recommended that a weighing rain gauge be employed to verify precipitation when $T_w$ is below 2 ℃.
At 2 ℃ or higher $T_w$, the DSD shape remained consistent across different QC methods. However, as the temperature dropped
below 2 ℃, Method 1, which defines the raindrop size interval as ±40% of the terminal velocity, indicated a comparatively
elevated number of drops measuring 2 mm or larger. In conditions where the temperature was below 1 ℃, the application of
Method 3 (Raupach et al., 2015) resulted in a notable and irregular distribution of number concentration for droplet
diameters ranging from 1 to 2 mm. These findings can be attributed to the QC method employed, which raises concerns




about the reliability of the DSD parameters derived from the altered shape of the DSD. Consequently, it is imperative to use
disdrometer data corresponding to environmental conditions with temperatures of 2 °C or higher when calculating DSD
parameters and DSD-based rain rates.
**Appendices**
Figures A1 and A2 present the findings from a comparative analysis of hourly accumulated precipitation intensity, derived
from observations using the 2DVD and two types of rain gauges (A1 represents the tipping-bucket type and A2 represents
the weighing type) under the specified $T_w$ interval conditions at 1 °C intervals. In each figure, the solid line denotes the
regression line correlating the precipitation intensities derived by applying each quality control (QC) method. The constants
and validation indices associated with the regression lines are listed in Tables A1 and A2, respectively. Tables A3 and A4
provide details regarding the diameter and velocity channels used to calculate the number concentration based on 2DVD data.
The channel information corresponded to the values employed in the PARSIVEL disdrometer data.
























(a) -5 ℃ ≤ T_w < -4 ℃          (b) -4 ℃ ≤ T_w < -3 ℃          (c) -3 ℃ ≤ T_w < -2 ℃

(d) -2 ℃ ≤ T_w < -1 ℃          (e) -1 ℃ ≤ T_w < 0 ℃          (f) 0 ℃ ≤ T_w < 1 ℃

(g) 1 ℃ ≤ T_w < 2 ℃          (h) 2 ℃ ≤ T_w < 3 ℃          (i) 3 ℃ ≤ T_w < 4 ℃

(j) 4 ℃ ≤ T_w < 5 ℃



**Figure A1: Precipitation intensity scatter plot based on tipping-bucket rain gauge and 2DVD observation data for each $T_w$ range. Each color in the scatter plot represents a filtering method.**

(a) -5 ℃ ≤ T_w < -4 ℃

(b) -4 ℃ ≤ T_w < -3 ℃

(c) -3 ℃ ≤ T_w < -2 ℃

(d) -2 ℃ ≤ T_w < -1 ℃

(e) -1 ℃ ≤ T_w < 0 ℃

(f) 0 ℃ ≤ T_w < 1 ℃

(g) 1 ℃ ≤ T_w < 2 ℃

(h) 2 ℃ ≤ T_w < 3 ℃

(i) 3 ℃ ≤ T_w < 4 ℃

(j) 4 ℃ ≤ T_w < 5 ℃



**Figure A2: Precipitation intensity scatter plot based on weighing rain gauge and 2DVD observation data for each $T_w$ range. Each color in the scatter plot represents a filtering method.**

**Table A1: Comparison of precipitation intensity based on tipping-bucket rain gauge and 2DVD observation data for each $T_w$ range.**

| | | T1 | T2 | T3 | T4 | T5 | T6 | T7 | T8 | T9 | T10 |
|---|---|---|---|---|---|---|---|---|---|---|---|
| Unfiltered | $a_1$ | 5.88 | 3.74 | 3.72 | 4.40 | 2.11 | 0.73 | 1.46 | 1.05 | 1.09 | 1.06 |
| | $a_0$ | 0.12 | 0.11 | 0.25 | 0.11 | 0.32 | 0.43 | 0.01 | 0.04 | 0.04 | 0.06 |
| | RMSE | 1.01 | 0.91 | 0.92 | 0.96 | 0.85 | 0.92 | 0.46 | 0.10 | 0.17 | 0.23 |
| | MAE | 0.44 | 0.47 | 0.56 | 0.45 | 0.45 | 0.40 | 0.16 | 0.06 | 0.10 | 0.16 |
| | MAPE | 48.71 | 55.32 | 60.15 | 50.58 | 50.86 | 43.25 | 34.45 | 15.68 | 13.54 | 21.14 |
| | CC | 0.97 | 0.97 | 0.93 | 0.95 | 0.74 | 0.23 | 0.85 | 0.99 | 0.99 | 0.99 |
| Method 1 | $a_1$ | 0.39 | 0.28 | 0.51 | 1.18 | 0.49 | 0.71 | 1.23 | 1.00 | 1.04 | 1.01 |
| | $a_0$ | 0.04 | 0.02 | 0.05 | 0.06 | 0.14 | 0.07 | 0.01 | 0.04 | 0.04 | 0.05 |
| | RMSE | 0.17 | 0.18 | 0.12 | 0.21 | 0.29 | 0.20 | 0.30 | 0.09 | 0.12 | 0.16 |
| | MAE | 0.08 | 0.07 | 0.05 | 0.11 | 0.15 | 0.11 | 0.10 | 0.06 | 0.07 | 0.12 |
| | MAPE | 28.57 | 56.67 | 27.24 | 34.01 | 37.07 | 56.39 | 31.27 | 17.96 | 12.04 | 16.83 |
| | CC | 0.45 | 0.75 | 0.78 | 0.79 | 0.55 | 0.66 | 0.89 | 0.99 | 1.00 | 0.99 |
| Method 2 | $a_1$ | 0.70 | 0.58 | 0.37 | 0.59 | 0.34 | 0.87 | 1.03 | 0.98 | 1.00 | 0.96 |
| | $a_0$ | 0.01 | 0.00 | 0.01 | 0.03 | 0.08 | 0.02 | 0.00 | 0.03 | 0.04 | 0.06 |
| | RMSE | 0.03 | 0.03 | 0.06 | 0.13 | 0.20 | 0.09 | 0.12 | 0.08 | 0.08 | 0.13 |
| | MAE | 0.01 | 0.01 | 0.04 | 0.07 | 0.10 | 0.06 | 0.05 | 0.05 | 0.05 | 0.09 |
| | MAPE | 5.56 | 11.11 | 23.81 | 57.78 | 41.67 | 26.56 | 26.79 | 12.78 | 10.54 | 13.44 |
| | CC | 0.98 | 0.94 | 0.60 | 0.67 | 0.57 | 0.91 | 0.97 | 0.99 | 1.00 | 1.00 |
| Method 3 | $a_1$ | 1.29 | 2.22 | 2.30 | 2.18 | 0.92 | 0.76 | 1.28 | 1.05 | 1.09 | 1.06 |
| | $a_0$ | 0.09 | 0.09 | 0.12 | 0.13 | 0.18 | 0.24 | 0.00 | 0.04 | 0.04 | 0.06 |
| | RMSE | 0.24 | 0.44 | 0.47 | 0.49 | 0.33 | 0.43 | 0.25 | 0.10 | 0.17 | 0.23 |
| | MAE | 0.11 | 0.26 | 0.28 | 0.26 | 0.18 | 0.23 | 0.10 | 0.06 | 0.10 | 0.16 |
| | MAPE | 38.75 | 49.04 | 51.11 | 47.07 | 45.58 | 46.25 | 25.58 | 15.68 | 13.54 | 21.14 |
| | CC | 0.76 | 0.99 | 0.94 | 0.87 | 0.73 | 0.47 | 0.94 | 0.99 | 0.99 | 0.99 |
| Method 1 (melted) | $a_1$ | 0.81 | 0.71 | 0.81 | 1.29 | 0.59 | 0.67 | 1.26 | 1.03 | 1.06 | 1.02 |
| | $a_0$ | 0.07 | 0.03 | 0.08 | 0.07 | 0.15 | 0.16 | 0.01 | 0.04 | 0.03 | 0.05 |
| | RMSE | 0.16 | 0.10 | 0.15 | 0.25 | 0.28 | 0.31 | 0.32 | 0.09 | 0.13 | 0.16 |
| | MAE | 0.06 | 0.05 | 0.09 | 0.12 | 0.16 | 0.16 | 0.11 | 0.06 | 0.08 | 0.12 |
| | MAPE | 26.33 | 21.39 | 34.06 | 31.87 | 42.73 | 38.50 | 31.64 | 18.03 | 12.37 | 16.51 |
| | CC | 0.76 | 0.97 | 0.85 | 0.82 | 0.66 | 0.51 | 0.88 | 0.99 | 1.00 | 0.99 |





|  |  |  |  |  |  |  |  |  |  |  |
|---|---|---|---|---|---|---|---|---|---|---|
|  | $a_1$ | 0.71 | 0.59 | 0.60 | 0.86 | 0.42 | 0.68 | 1.08 | 0.99 | 1.02 | 0.98 |
|  | $a_0$ | 0.02 | 0.02 | 0.06 | 0.06 | 0.10 | 0.10 | 0.00 | 0.03 | 0.04 | 0.05 |
| Method 2 (melted) | RMSE | 0.09 | 0.14 | 0.15 | 0.17 | 0.25 | 0.21 | 0.14 | 0.08 | 0.09 | 0.13 |
|  | MAE | 0.05 | 0.07 | 0.08 | 0.09 | 0.13 | 0.12 | 0.06 | 0.05 | 0.05 | 0.10 |
|  | MAPE | 17.78 | 23.33 | 27.88 | 31.87 | 45.37 | 44.88 | 27.62 | 12.84 | 11.00 | 15.80 |
|  | CC | 0.93 | 0.98 | 0.84 | 0.79 | 0.67 | 0.65 | 0.96 | 0.99 | 1.00 | 1.00 |
|  | $a_1$ | 1.54 | 2.33 | 2.41 | 2.25 | 0.99 | 0.77 | 1.29 | 1.05 | 1.09 | 1.06 |
|  | $a_0$ | 0.10 | 0.08 | 0.14 | 0.13 | 0.19 | 0.26 | 0.00 | 0.04 | 0.04 | 0.06 |
| Method 3 (melted) | RMSE | 0.27 | 0.46 | 0.51 | 0.51 | 0.35 | 0.46 | 0.26 | 0.10 | 0.17 | 0.23 |
|  | MAE | 0.13 | 0.27 | 0.30 | 0.28 | 0.20 | 0.25 | 0.11 | 0.06 | 0.10 | 0.16 |
|  | MAPE | 40.50 | 49.24 | 50.27 | 47.38 | 46.46 | 45.30 | 28.44 | 15.68 | 13.54 | 21.14 |
|  | CC | 0.79 | 0.99 | 0.93 | 0.87 | 0.74 | 0.45 | 0.93 | 0.99 | 0.99 | 0.99 |


**Table A2: Comparison of precipitation intensity based on weighing rain gauge and 2DVD observation data for each $T_w$ range.**

|  |  | T1 | T2 | T3 | T4 | T5 | T6 | T7 | T8 | T9 | T10 |
|---|---|---|---|---|---|---|---|---|---|---|---|
|  | $a_1$ | 7.36 | 3.39 | 3.98 | 3.86 | 2.25 | 2.15 | 1.37 | 1.01 | 1.03 | 1.03 |
|  | $a_0$ | 0.09 | 0.12 | 0.27 | 0.09 | 0.24 | 0.06 | 0.04 | 0.04 | 0.02 | 0.01 |
| Unfiltered | RMSE | 1.04 | 0.88 | 0.95 | 0.93 | 0.76 | 0.64 | 0.33 | 0.14 | 0.12 | 0.17 |
|  | MAE | 0.44 | 0.46 | 0.57 | 0.43 | 0.43 | 0.31 | 0.15 | 0.08 | 0.07 | 0.12 |
|  | MAPE | 49.76 | 55.42 | 60.80 | 48.81 | 53.44 | 41.61 | 41.39 | 22.03 | 13.07 | 19.05 |
|  | CC | 0.98 | 0.97 | 0.90 | 0.93 | 0.88 | 0.89 | 0.95 | 0.98 | 0.99 | 0.99 |
|  | $a_1$ | 0.61 | 0.26 | 0.48 | 0.93 | 0.48 | 0.61 | 1.11 | 0.97 | 0.98 | 0.98 |
|  | $a_0$ | 0.02 | 0.03 | 0.06 | 0.05 | 0.09 | 0.05 | 0.04 | 0.05 | 0.02 | 0.01 |
| Method 1 | RMSE | 0.13 | 0.17 | 0.13 | 0.19 | 0.28 | 0.20 | 0.20 | 0.13 | 0.10 | 0.14 |
|  | MAE | 0.06 | 0.07 | 0.07 | 0.12 | 0.13 | 0.11 | 0.10 | 0.08 | 0.07 | 0.10 |
|  | MAPE | 25.71 | 63.33 | 41.35 | 41.49 | 36.12 | 79.17 | 38.21 | 22.17 | 14.21 | 21.89 |
|  | CC | 0.70 | 0.65 | 0.73 | 0.80 | 0.79 | 0.71 | 0.96 | 0.98 | 0.99 | 1.00 |
|  | $a_1$ | 0.53 | 0.58 | 0.21 | 0.46 | 0.28 | 0.80 | 0.88 | 0.95 | 0.94 | 0.93 |
|  | $a_0$ | 0.01 | 0.00 | 0.01 | 0.03 | 0.06 | 0.01 | 0.05 | 0.03 | 0.03 | 0.02 |
| Method 2 | RMSE | 0.07 | 0.03 | 0.09 | 0.18 | 0.31 | 0.10 | 0.11 | 0.12 | 0.11 | 0.17 |
|  | MAE | 0.02 | 0.01 | 0.06 | 0.11 | 0.15 | 0.07 | 0.07 | 0.07 | 0.07 | 0.12 |
|  | MAPE | 22.22 | 11.11 | 28.57 | 85.56 | 55.43 | 27.34 | 35.56 | 18.32 | 14.12 | 20.69 |
|  | CC | 0.82 | 0.94 | 0.45 | 0.75 | 0.78 | 0.92 | 0.98 | 0.98 | 1.00 | 1.00 |
| Method 3 | $a_1$ | 1.91 | 2.01 | 2.33 | 2.02 | 0.92 | 1.04 | 1.12 | 1.01 | 1.03 | 1.03 |



| | | | | | | | | | | | |
|---|---|---|---|---|---|---|---|---|---|---|---|
| | $a_0$ | 0.07 | 0.10 | 0.15 | 0.10 | 0.15 | 0.10 | 0.05 | 0.04 | 0.02 | 0.01 |
| | RMSE | 0.22 | 0.41 | 0.52 | 0.45 | 0.29 | 0.21 | 0.16 | 0.14 | 0.12 | 0.17 |
| | MAE | 0.12 | 0.25 | 0.29 | 0.24 | 0.16 | 0.13 | 0.09 | 0.08 | 0.07 | 0.12 |
| | MAPE | 42.13 | 49.58 | 52.32 | 44.83 | 46.50 | 37.92 | 34.21 | 22.03 | 13.07 | 19.05 |
| | CC | 0.91 | 0.98 | 0.86 | 0.90 | 0.80 | 0.90 | 0.98 | 0.98 | 0.99 | 0.99 |
| Method 1 (melted) | $a_1$ | 1.16 | 0.65 | 0.89 | 1.12 | 0.57 | 0.79 | 1.15 | 0.99 | 0.99 | 0.98 |
| | $a_0$ | 0.05 | 0.03 | 0.09 | 0.06 | 0.12 | 0.06 | 0.04 | 0.04 | 0.02 | 0.01 |
| | RMSE | 0.11 | 0.13 | 0.16 | 0.23 | 0.26 | 0.16 | 0.21 | 0.13 | 0.10 | 0.13 |
| | MAE | 0.06 | 0.08 | 0.10 | 0.11 | 0.15 | 0.10 | 0.09 | 0.08 | 0.06 | 0.10 |
| | MAPE | 30.25 | 31.05 | 36.50 | 29.44 | 49.92 | 32.10 | 37.80 | 22.22 | 13.91 | 21.50 |
| | CC | 0.90 | 0.97 | 0.82 | 0.82 | 0.77 | 0.89 | 0.96 | 0.98 | 1.00 | 1.00 |
| Method 2 (melted) | $a_1$ | 0.91 | 0.53 | 0.65 | 0.74 | 0.38 | 0.61 | 0.93 | 0.96 | 0.95 | 0.95 |
| | $a_0$ | 0.01 | 0.02 | 0.07 | 0.05 | 0.09 | 0.04 | 0.05 | 0.04 | 0.02 | 0.01 |
| | RMSE | 0.04 | 0.18 | 0.14 | 0.17 | 0.29 | 0.18 | 0.09 | 0.12 | 0.11 | 0.15 |
| | MAE | 0.01 | 0.10 | 0.09 | 0.11 | 0.14 | 0.10 | 0.06 | 0.07 | 0.07 | 0.11 |
| | MAPE | 8.89 | 35.56 | 34.62 | 42.75 | 58.79 | 40.35 | 34.42 | 17.78 | 14.24 | 22.16 |
| | CC | 0.98 | 0.98 | 0.79 | 0.82 | 0.77 | 0.87 | 0.99 | 0.98 | 1.00 | 1.00 |
| Method 3 (melted) | $a_1$ | 2.24 | 2.10 | 2.47 | 2.08 | 0.99 | 1.14 | 1.13 | 1.01 | 1.03 | 1.03 |
| | $a_0$ | 0.07 | 0.10 | 0.16 | 0.11 | 0.16 | 0.10 | 0.05 | 0.04 | 0.02 | 0.01 |
| | RMSE | 0.27 | 0.44 | 0.55 | 0.48 | 0.30 | 0.22 | 0.16 | 0.14 | 0.12 | 0.17 |
| | MAE | 0.14 | 0.26 | 0.32 | 0.25 | 0.17 | 0.14 | 0.09 | 0.08 | 0.07 | 0.12 |
| | MAPE | 43.48 | 49.81 | 51.47 | 45.14 | 46.83 | 38.15 | 37.14 | 22.03 | 13.07 | 19.05 |
| | CC | 0.93 | 0.98 | 0.86 | 0.89 | 0.82 | 0.92 | 0.98 | 0.98 | 0.99 | 0.99 |














**Table A3: Diameter channel information of the PARSIVEL disdrometer.**

| Channel number | Mid-value of channel (mm) | Diameter spread (mm) | | Channel number | Mid-value of channel (mm) | Diameter spread (mm) |
|---|---|---|---|---|---|---|
| 1 | 0.062 | 0.125 | | 17 | 3.250 | 0.500 |
| 2 | 0.187 | 0.125 | | 18 | 3.750 | 0.500 |
| 3 | 0.312 | 0.125 | | 19 | 4.250 | 0.500 |
| 4 | 0.437 | 0.125 | | 20 | 4.750 | 0.500 |
| 5 | 0.562 | 0.125 | | 21 | 5.500 | 1.000 |
| 6 | 0.687 | 0.125 | | 22 | 6.500 | 1.000 |
| 7 | 0.812 | 0.125 | | 23 | 7.500 | 1.000 |
| 8 | 0.937 | 0.125 | | 24 | 8.500 | 1.000 |
| 9 | 1.062 | 0.125 | | 25 | 9.500 | 1.000 |
| 10 | 1.187 | 0.125 | | 26 | 11.000 | 2.000 |
| 11 | 1.375 | 0.250 | | 27 | 13.000 | 2.000 |
| 12 | 1.625 | 0.250 | | 28 | 15.000 | 2.000 |
| 13 | 1.875 | 0.250 | | 29 | 17.000 | 2.000 |
| 14 | 2.125 | 0.250 | | 30 | 19.000 | 2.000 |
| 15 | 2.375 | 0.250 | | 31 | 21.500 | 3.000 |
| 16 | 2.750 | 0.500 | | 32 | 24.500 | 3.000 |





**Table A4: Velocity channel information of the PARSIVEL disdrometer.**

| Channel number | Mid-value of channel (mm) | Velocity spread (mm) | | Channel number | Mid-value of channel (mm) | Velocity spread (mm) |
|---|---|---|---|---|---|---|
| 1 | 0.050 | 0.100 | | 17 | 2.600 | 0.400 |
| 2 | 0.150 | 0.100 | | 18 | 3.000 | 0.400 |
| 3 | 0.250 | 0.100 | | 19 | 3.400 | 0.400 |
| 4 | 0.350 | 0.100 | | 20 | 3.800 | 0.400 |
| 5 | 0.450 | 0.100 | | 21 | 4.400 | 0.800 |
| 6 | 0.550 | 0.100 | | 22 | 5.200 | 0.800 |
| 7 | 0.650 | 0.100 | | 23 | 6.000 | 0.800 |
| 8 | 0.750 | 0.100 | | 24 | 6.800 | 0.800 |
| 9 | 0.850 | 0.100 | | 25 | 7.600 | 0.800 |
| 10 | 0.950 | 0.100 | | 26 | 8.800 | 1.600 |
| 11 | 1.100 | 0.200 | | 27 | 10.400 | 1.600 |
| 12 | 1.300 | 0.200 | | 28 | 12.000 | 1.600 |
| 13 | 1.500 | 0.200 | | 29 | 13.600 | 1.600 |
| 14 | 1.700 | 0.200 | | 30 | 15.200 | 1.600 |
| 15 | 1.900 | 0.200 | | 31 | 17.600 | 3.200 |
| 16 | 2.200 | 0.400 | | 32 | 20.800 | 3.200 |



**Author contributions**
HJK and CJ, conceptualized the project. JB did the data curation and formal analysis. HJK and SHS did the analysis and
interpretation. HJK and JB led the investigation. HJK prepared the original draft, and SHS and CJ reviewed and edited the
paper. All authors have read and agreed to the published version of the paper.
**Acknowledgements**
This study was supported by the Basic Science Research Program through the National Research Foundation of Korea
(NRF), funded by the Ministry of Education (RS-2022-NR071182).

**Competing Interest**
The contact author has declared that none of the authors has any competing interests.



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
