# Peer review of "Assessment of Disdrometer Data Quality Control Methods for Precipitation Measurements Based on Wet-Bulb Temperature"

_EGUsphere, 2025_

## Author Comment (AC1)

**Assessment of Disdrometer Data Quality Control Methods for Precipitation Measurements Based on Wet-Bulb Temperature**

**By H. J. Kim *et al*.**

Reply to the referees' comments

In the following, the comments made by the referees appear in black, while our replies are in red, and the proposed modified text in the typescript is in blue.

**Referee #1 comments**

This paper represents a significant contribution to the field by improving the accuracy and reliability of precipitation data through the evaluation of quality control methods for precipitation measurement instruments, with a particular emphasis on the impact of wet-bulb temperature. The research is of considerable value. But revisions are required before publication.

Authors are grateful for reviewer's interest in this study and the many helpful suggestions for improving this manuscript. Replies to each major comments and minor comments are listed below.

**Major comments**

1. Line 98-101: In this part of the discussion, it is essential to include relevant references. Additionally, it is suggested that the authors polish the language of the entire text to improve its quality.

We agree. In order to reinforce the basis for the content regarding the relationship between the diameter and fall velocity of raindrops, as you mentioned, we added references to the sentence in question. We added a reference that explains that as the diameter of raindrops increases, their fall velocity also increases, and when the fall velocity increases to the point where buoyancy and gravity balance each other, the fall velocity reaches terminal velocity.

♧ Page 4, line 97-101

"QC approaches for disdrometer data primarily rely on the falling velocity of raindrops. In the absence of a substantial wind influence or particle collisions during descent, the fall velocity of a raindrop tends to increase with its diameter, ultimately reaching a terminal velocity. Terminal velocity is achieved when the forces of air resistance and gravitational pull are in equilibrium, resulting in no further particle acceleration (Wang and Pruppacher, 1977; Ong et al., 2021)."

♧ Page 41, line 744-745

"Ong, C. R., Miura, H., & Koike, M.: The terminal velocity of axisymmetric cloud drops and raindrops evaluated by the immersed boundary method. J. Atmos Sci., 78(4), 1129–1146. https://doi.org/10.1175/JAS-D-20-0161.1, 2021."

♧ Page 42, line 782-783

"Wang, P. K., & Pruppacher, H. R.: Acceleration to terminal velocity of cloud and raindrops. J. Appl. Meteorol. Clim., 16(3), 275–280. https://doi.org/10.1175/1520-0450(1977)016<0275:ATTVOC>2.0.CO;2, 1977."

2. Line 238-240 : Given that the QC methods for the disdrometer were specifically designed to address rainfall, the variable Tw was employed to differentiate between rainfall and snowfall, thereby facilitating the verification of rainfall timing. I would like to know what exactly is meant by "rainfall timing." Does it refer to the start time of the rainfall or its duration? How is it verified? This issue is not mentioned in the subsequent discussion.

Thank you for your detailed comments. The "rainfall timing" mentioned in the sentence refers to the time when rainfall was observed using disdrometer such as 2DVD and PARSIVEL. Since previous studies have mentioned that differences in the distribution of rainfall and snowfall appear depending on Tw, this study conducted a detailed analysis of whether changes in precipitation characteristics (fall velocity) based on disdrometer observations appear depending on the $T_w$ variable. To improve readability, the sentence has been revised as follows.

♧ Page 10, line 239-241

"Given that the QC methods for the disdrometer were specifically designed to address rainfall, the variable Tw was employed to differentiate between rainfall and snowfall, thereby facilitating the verification of precipitation type."

3. Figure 9 may cause some confusion due to the presence of two dashed lines in the same color. It is recommended to make appropriate modifications to enhance the clarity and readability of the figure.

We appreciate your constructive feedback. In response to your suggestion, we have revised the lines depicting the fall velocity intervals of raindrops for each quality control method in Figure 9, converting them to solid lines to enhance clarity. Furthermore, we have modified the colors better to differentiate these lines from the observed fall velocities.

[Figure]

**Figure 9: Distribution of fall speed by diameter under conditions of $T_w \geq 5$ °C, and effective fall velocity of raindrops by pre-processing methods (The red (method 1), orange (method 2), and purple (method 3) solid lines represent the effective velocity of raindrop applied to each QC method).**

4. It is recommended to include a discussion on the limitations of the methods used in this paper. When discussing the effectiveness of the QC method at different wet bulb temperatures, should it also be influenced and constrained by factors such as the measurement location?

We agree. Previous studies (Ding et al., 2014; Hasager et al., 2020) have mentioned results on the classification of precipitation types (rain and snow) based on wet-bulb temperature. This study focuses on interpreting results obtained using ground-based disdrometers, rain gauges, and AWS data. Characteristics such as hydrometeor types are primarily influenced by environmental factors like temperature, making it challenging to determine changes in precipitation types solely based on the observation location of the instrument. Therefore, when conducting quantitative analyses using rainfall data from in-situ observation instruments, such as disdrometers and rain gauges, it is necessary to consider environmental variables, including temperature, in the analysis.

- Ding, B., Yang, K., Qin, J., Wang, L., Chen, Y., & He, X. (2014). The dependence of precipitation types on surface elevation and meteorological conditions and its parameterization. Journal of hydrology, 513, 154-163.

- Hasager, C., Vejen, F., Bech, J. I., Skrzypiński, W. R., Tilg, A. M., & Nielsen, M. (2020). Assessment of the rain and wind climate with focus on wind turbine blade leading edge erosion rate and expected lifetime in Danish Seas. Renewable Energy, 149, 91-102.

**Minor comments**

1. The units of the vertical axis in Figures 13 and 14 are inconsistent with the units described in the text. The author should carefully check this.

Thank you for your kind feedback. We have revised the text to make it easier for readers to understand the contents of the figures. In addition, we have clearly checked and reviewed the units of the vertical axis in Figures 13 and 14. The revisions are as follows.

♣ Page 20, line 403-414

"Precipitation measurements obtained from the disdrometer were derived from raindrop (or snow particle) accumulation. The quantitative errors associated with these precipitation measurements were assessed by comparing the filter rates of raindrops (or snow particles) using the QC method. Figure 13 shows the filter ratios corresponding to the Tw range and channel diameter. The two methods, Method 1 and Method 2, exhibit differences in the range of removal velocities based on particle diameter (see Fig. 9); specifically, Method 2 encompasses a broader spectrum of raindrop sizes compared to Method 1, leading to an increased filter rate when the Tw is below 0 ℃. Notably, the filter rate for Method 2 surpasses that of Method 1 at temperatures lower than -2 ℃. Conversely, Method 3 did not allow the removal of particles smaller than 2 mm (as indicated in CH 14), regardless of their low fall velocity, resulting in a consistent filter rate of 0%, irrespective of variations in $T_w$. This suggests that the number of particles smaller than 2 mm may be greater in Methods 1 and 2. Furthermore, the filter rate was lower when snow particles were assumed to have melted than when they had not melted. Nonetheless, for particles with a diameter of 1 mm or less, the filter rate ranged from approximately 10% to 30% when $T_w$ exceeded 1 ℃, which appears to be attributable to the removal of particles exhibiting a fall velocity that exceeds the raindrops."

---

## Author Comment (AC2)

Assessment of Disdrometer Data Quality Control Methods for Precipitation Measurements Based on Wet-Bulb Temperature

By H. J. Kim et al.

Reply to the referees' comments

In the following, the comments made by the referees appear in black, while our replies are in red, and the proposed modified text in the typescript is in blue.

**Referee #2 comments**

The manuscript evaluates three data quality control methods for disdrometer measurements based on wet-bulb temperature. This work is valuable as it may promote the application of disdrometer observations across diverse types of precipitation. In current version, imprecise expressions are present throughout the manuscript, particularly in the descriptions of the figures, which hinders general readers from clearly understanding the study. There is still considerable room for improvement in the scientific expression. In addition, the rational for selecting the three quality control methods needs to be further justified, as their comparison does not reveal significant differences. Several specific comments are provided below for possible improvement.

Authors are grateful for reviewer's interest in this study and the many helpful suggestions for improving this manuscript. Replies to each major comments and minor comments are listed below.

1. The current Title may be refined to more clearly reflect the central focus of the manuscript.

We appreciate your constructive feedback. The title has been revised as follows to more clearly convey the content of this study.

♣ Page 1, line 1-2

Validation of Rainfall Data Analysis Observed by Using Disdrometer under Wet-Bulb Temperature Conditions

2. The Introduction section occasionally presents results that should be placed in later sections, and lacks appropriate references. For examples, in Lines 117-119: "the authors noted a tendency for PARSIVEL to overestimate the number of small droplets measuring between 0.2 and 0.4 mm and larger particles measuring 2.4 mm or more. Furthermore, the measured fall velocity of larger droplets was lower than the actual terminal velocity". Any appropriate reference? In Lines 129-131: "Given the diverse shapes and fall speeds of snow particles, the mixing of raindrops and snow during precipitation events may lead to an underestimation of errors when applying conventional disdrometer QC methods." Any references? Similar issues exist elsewhere in the manuscript. For example, in Lines 181: "numerous studies", any citations?

Thank you for your detailed comments. According to Raupach et al. (2015), the PARSIVEL disdrometer overestimates the number of small drops with diameters of 0.7 mm or less, and specifically overestimates in all channel sections up to 4 mm under rain rate conditions weaker than 0.1 mm h-1. Furthermore, Tokay et al. (2013) also noted that the PARSIVEL disdrometer overestimates particle counts for particles larger

than 2.44 mm. Furthermore, channels 1 and 2 of the PARSIVEL disdrometer's diameter channels do not collect data valid for analysis due to signal-to-noise issues. Therefore, the effective minimum diameter can be considered 0.2 mm. According to the findings of Raupach et al. (2015), PARSIVEL shows lower particle fall speeds compared to the 2DVD (Two-dimensional Video Disdrometer).

- [...] between the values in the P (i) curve and the rain intensity. The most notable feature of Fig. 8 is that the numbers of small drops (under about 0.7 mm) were overestimated by the Parsivel. [...], For low rain rates, below 1 mm  $h^{-1}$ , the Parsivel overestimated drop counts in all classes up to 4 mm. (Raupach et al., 2015)
- [...] Tokay et al. (2013) found that Parsivel disdrometers were less sensitive to small drops than the 2DVD, and that they overestimated the numbers of drops over 2.44 mm in diameter, [...]. (Raupach et al., 2015)
- \*\* Tokay, A., Petersen, W. A., Gatlin, P., & Wingo, M. (2013). Comparison of raindrop size distribution measurements by collocated disdrometers. Journal of Atmospheric and Oceanic Technology, 30(8), 1672-1690.)

Fig. a1. Sum of raw drop occurrences per Parsivel class, for the 2012 and 2013 campaigns. Parsivel counts are summed at stations Pradel 1 (for 2012) and Pradel Grainage (for 2013). The filtered areas are overlaid in grey. The black line is the expected terminal drop velocity calculated by Beard (1976). Drop counts are specified by colour on a log scale. (Raupach et al., 2015)

While liquid droplets such as raindrops can be assumed to have a fixed density, solid particles like snow exhibit density variations relative to their diameter and possess a lower density compared to raindrops. Consequently, the fall velocity of snow particles is lower than that of raindrops. Applying the QC method

designed for raindrops to snow particles may therefore yield underestimated results (Fehlmann et al., 2020; Lachapelle et al., 2024).

Fig. a2. Example of the classification algorithm developed in this study during a transition from rain to snowfall (17 February 2018, 17:00 to 23:00 UTC). After a plausibility check, each hydrometeor detected by the two-dimensional video disdrometer is classified as one of five precipitation types (hail, rain, melting snow, graupel, snow). This classification is based on empirical relationships between particle diameter and fall velocity. (Fehlmann et al., 2020)

Fig. a3. Relationship between snow particle density and mean particle diameter based on 1 min observations during the first year of measurements. Snowfall events are identified based on the recorded dominant precipitation type by the Thies disdrometer. Snow particle density is then calculated by comparing the precipitation volume measured by the two-dimensional video disdrometer (2DVD) and precipitation mass measured by the OTT pluviometer and is related to mean particle diameter as measured by the 2DVD. The fitted curve is used to translate particle size distribution into snowfall intensities during the second year of measurements. Note that the corresponding relationship established by Brandes et al. (2007) is shown as a reference. (Fehlmann et al., 2020)

Fig. a4. Representation of laser-optical disdrometer measurements during rain episodes observed on (a) 1200–1500 UTC 19 Nov 2019, (b) 0800–1700 UTC 12 Jan 2020, (c) 1000–1100 UTC 27 Feb 2020, and (d) 0400–0800 UTC 23 Nov 2020. The solid, colored lines are theoretical fall speed curves for FZRA and RA, PL, hail, SN, GS, and SN. (Lachapelle et al., 2024)

- \*\* Fehlmann, M., Rohrer, M., von Lerber, A., & Stoffel, M. (2020). Automated precipitation monitoring with the Thies disdrometer: Biases and ways for improvement. Atmospheric Measurement Techniques Discussions, 2020, 1-31.
- \*\* Lachapelle, M., Thompson, H. D., Leroux, N. R., & Thériault, J. M. (2024). Measuring ice pellets and refrozen wet snow using a laser-optical disdrometer. Journal of Applied Meteorology and Climatology, 63(1), 65-84.

The revisions with the added references are as follows.

**♣ Page 8, line 181-182**

"A common QC approach for disdrometer data involves excluding non-meteorological data by analyzing fall velocity. In numerous studies (Kruger and Krajewski, 2002; Jaffrain and Berne, 2011; Raupach and Berne, 2015; Kim et al., 2019), [...]"

- \*\* Kruger, A., & Krajewski, W. F. (2002). Two-dimensional video disdrometer: A description. Journal of Atmospheric and Oceanic Technology, 19(5), 602-617.
- \*\* Jaffrain, J., & Berne, A. (2011). Experimental quantification of the sampling uncertainty associated with measurements from PARSIVEL disdrometers. Journal of Hydrometeorology, 12(3), 352-370.
- \*\* Raupach, T. H., & Berne, A. (2015). Correction of raindrop size distributions measured by Parsivel disdrometers, using a two-dimensional video disdrometer as a reference. Atmospheric Measurement

**Techniques, 8(1), 343-365**

- \*\* Kim, H. J., Lee, K. O., You, C. H., Uyeda, H., & Lee, D. I. (2019). Microphysical characteristics of a convective precipitation system observed on July 04, 2012, over Mt. Halla in South Korea. Atmospheric Research, 222, 74-87.
- 3. The manuscript does not clearly describe the conventional QC methods. Please clarify what these conventional approaches are and explicitly discuss how they differ from the three QC methods selected in this study.

The existing QC methods mentioned in this study are based on setting effective ranges using terminal velocity values for raindrops of different diameters, and they differ in how these ranges are defined. These methods are applicable because they consider the water liquid density to be fixed for raindrops, and the terminal velocity varies with diameter. Methods 1 and 2 set the valid range at ±40% and ±60% of the terminal velocity, respectively. Method 3, however, sets a fixed range rather than a percentage of the terminal velocity value. It considers all small drops smaller than 2 mm to be valid if they have a fall velocity lower than the terminal velocity.

**♣ Page 8, line 188-190**

"[...] predominantly adopted a setting constant of 0.4 (40%) during data processing. Studies that employed PARSIVEL data for analysis frequently applied a setting constant of 0.6, accounting for 60% of the cases [...]"

**♠ Page 8, line 195-197**

- "[...] The fall velocity filtering technique employed for the 2DVD and PARSIVEL data involved the exclusion of particles exhibiting a terminal velocity exceeding 4 m s-1, as shown in Eq. (2), those with a fall velocity below 3 m s-1 [...]"
- 4. Section 3.1: The manuscript should report the proportion of disdrometer data removed by each QC method to allow for a clearer comparison of their performance. Furthermore, please clarify whether data associated with solid meteorological particles (e.g., snow, as indicated in Lines 204–205) may be removed by these methods.

Thank you for your kind feedback. The removal rates of solid meteorological particles before and after QC are shown in Fig. 13. These results indicate that the removal rate increases when the  $T_w$  condition is 1°C or lower, and when  $T_w$  drops below -2°C, the removal rate of particles increases to over 90%.

Figure 13: Particle filter ratio by diameter channel for  $T_w$  according to the pre-processing method based on falling velocity.

5. In Equation (5), is V(D) used to calculate Videal, i.e., terminal velocity? Please clarify.

We agree. In the expression in Equation 5, V(D) represents the terminal velocity value  $V_{ideal}$ , and the expression in the manuscript has also been revised as follows.

♣ Page 8, Equation (5)

$$V_{ideal}(D) = 9.65 - 10.3 \exp(-0.6D)$$
 (5)

6. Is the estimation of Tw from Tair and RH in Equation (15) applicable when Tair

Fig. a5. Isopleths of Tw (thick black curves) vs RH% and T, found from Tw Equation. The valid range is enclosed by a dashed line, and the valid pressure is 101.325 kPa. The gray curves associated with each Tw are for P 5 80 kPa (thinner lines) and P 5 60 kPa (thinnest lines, located farther away from each black line). These gray curves [not found from Tw Eq.)] are useful for estimating the error if Tw Eq. is applied to pressures that are not equal to 101.325 kPa.

\*\* Stull, R. (2011). Wet-bulb temperature from relative humidity and air temperature. Journal of applied meteorology and climatology, 50(11), 2267-2269.)

7. Figures 5-6: Is Rainfall[Gauge] the same as Rainfall[TG]? It would be better to keep consistent terminology. It's not easy to find the effect of QC in Figures 5-6. Please clarify the explicit differences between Methods 1 and 2 (Fig. 5b and 5c), or even Method 3 (Fig. 5d), and indicate whether these differences are significant? Including the number of datapoints in each panel would improve clarity and aid interpretation. In Line 256, the term "overestimate" is used—please clarify whether this applies to the comparison between Figure 6a and 6d as well. Significant?

We appreciate your constructive feedback. To improve the readability of the result figures, the figures were modified as follows. These results aim to assess the validity of each QC method for rainfall cases. As shown in Fig. 9, the rainfall cases fall within the valid range of the QC methods, indicating that they do not exhibit significant differences. Particularly for rainfall, QC methods based on fall velocity exhibit low error for large diameters ( $\geq 3$  mm) when observation conditions are well-controlled and non-meteorological values, such as leaves, are not detected. Differences in QC primarily occur for small diameters ( $\leq 1$  mm). As shown in Fig. 6, the Unfiltered case and Method 3 yield very similar results, indicating that raindrops possess a valid fall velocity. The relative underestimation observed in Methods 1 and 2 can be attributed to the partial removal of small drops. Although the Unfiltered results are relatively overestimated compared to those using QC methods, the RMSE for all methods in Figures 5-6 was less than approximately 1 mm, and the correlation

**coefficient exceeded 0.98. This indicates the high validity of the data.**

**♣ Page 11, line 258-262**

This discrepancy in the overestimation of the 2DVD data can be attributed to variations in the conditions under which particles are eliminated, which is contingent on the specific QC method employed. Following the application of the QC methods, the mean absolute percentage error (MAPE) demonstrated an overall reduction compared with the raw data, suggesting that all QC methods possess quantitative reliability for rainfall data, with a maximum reduction of approximately 2.1%.

**♠ Page 15, line 307-310**

The central value of the fall velocity is consistent with the terminal velocity. This is within the range of fall velocities for raindrops, as established by the three different QC methods based on the fall velocity. It is important to note that precipitation particles (drops) may experience variations in their fall velocities owing to factors such as wind influence or collisions with obstacles during descent.

Figure 5: Comparison of rainfall observed using the tipping-bucket rain gauge and 2DVD when  $T_w \ge 5$  °C ((a) Unfiltered, (b) Method 1, (c) Method 2, (d) Method 3).  $R_{2DVD}$  and  $R_{TG}$  denote the rainfall obtained from the 2DVD and a tipping-bucket rain gauge, respectively.

Figure 6: Comparison of rainfall observed using the weighing rain gauge and 2DVD when  $T_w \ge 5$  °C ((a) Unfiltered, (b) Method 1, (c) Method 2, (d) Method 3). RwG denotes the rainfall obtained from a weighing rain gauge.

8. Many expressions throughout the manuscript lack rigor. For examples, in Lines 279-280: "However, as the temperature exceeded 0  $^{\circ}$ C, the fall velocity for CH 4 to 18 increased under  $T_{w}$  conditions, while the fall velocity for CH 19 to 23 increased under  $T_{air}$  conditions? (Fig. 7(a-b))". Can find this result from 7a and 7b? It looks comparable between Figure 7a and 7b. In Lines 280-282: "Notably, when the temperature rose above 1  $^{\circ}$ C, there was a notable increase in fall velocity for CH 4 or larger? under Tw conditions, the distribution approached the terminal velocity of raindrops for CH 4 to 13?". The statement "Under  $T_{air}$  conditions, the fall velocity increased when temperatures were below 1 $^{\circ}$ C" is unclear. Please clarify how this statement can be determined from the presented data or figure?

Thank you for your detailed comments. The difference in fall velocity variation for  $T_w$  and  $T_{air}$  conditions can be clearly observed in Fig. 7(a-b). When  $T_w$  was below 0 °C, the upper 75% value of fall velocity was less than 2 m s-1. However, as Tw increased above 0°C, fall velocity increased to approximately 1 m s-1 or higher in the CH4–15 diameter range. Particularly in the CH8–11 range, the upper 75% value exceeded 3 m s-1. Specifically, up to CH13, fall velocity gradually increased with diameter, reaching large values exceeding 6 m s-1. Conversely, under  $T_{air}$  conditions, the upper 75% fall velocity values for the CH1–15 range were 2 m s-1 or less in the 0–1°C range. Under  $T_{air}$  conditions, the fall velocity increased when the temperature was above 1°C. However, it still differed from the terminal velocity of rainfall and exhibited a lower fall velocity than under  $T_w$  conditions. For a clearer explanation, it has been revised as follows.

♣ Page 13, line 282-290

However, as the temperature exceeded 0 °C, the fall velocity for CH 4 to 18 increased under Tw conditions,

while under the  $T_{air}$  condition, it exhibited values similar to those observed at temperatures below 0 °C (Fig. 7(a-b)). When  $T_w$  was below 0 °C, the upper 75% value of fall velocity was less than 2 m s-1. However, as Tw increased above 0 °C, fall velocity increased to approximately 1 m s-1 or higher in the CH4–15 diameter range. Particularly in the CH8–11 range, the upper 75% value exceeded 3 m s-1. Specifically, up to CH13, the fall velocity gradually increased with diameter, reaching large values exceeding 6 m s-1. Conversely, under  $T_{air}$  conditions, the upper 75% fall velocity values for the CH1–15 range were 2 m s-1 or less in the 0–1 °C range. Under  $T_{air}$  conditions, the fall velocity increased when the temperature was above 1 °C.